# VLA-4 suppression by senescence signals regulates meningeal immunity and leptomeningeal metastasis

Jiaqian Li[1,2†], Di Huang[1,2†], Bingxi Lei[3†], Jingying Huang[1,2], Linbing Yang[1,2], Man Nie[4], Shicheng Su[1,2,5*], Qiyi Zhao[6,7*], Ying Wang[1,2*]

[1]Guangdong Provincial Key Laboratory of Malignant Tumor Epigenetics and Gene Regulation, Medical Research Center, Sun Yat-Sen Memorial Hospital, Sun Yat-Sen University, Guangzhou, China; [2]Breast Tumor Center, Sun Yat-Sen Memorial Hospital, Sun Yat-Sen University, Guangzhou, China; [3]Department of Neurosurgery, Sun Yat-sen University, Guangzhou, China; [4]Department of Medical Oncology, State Key Laboratory of Oncology in South China, Collaborative Innovation Center for Cancer Medicine, Sun Yat-sen University Cancer Center, Guangzhou, China; [5]Department of Immunology, Zhongshan School of Medicine, Sun Yat-Sen University, Guangzhou, China; [6]Department of Infectious Diseases, Third Affiliated Hospital of Sun Yat-sen University, Guangzhou, China; [7]Guangdong Provincial Key Laboratory of Liver Disease Research, the Third Affiliated Hospital, Sun Yat-Sen University, Guangzhou, China

**\*For correspondence:**
sushch@mail.sysu.edu.cn (SS);
zhaoqyi@mail.sysu.edu.cn (QZ);
wangy556@mail.sysu.edu.cn
(YW)

†These authors contributed
equally to this work

**Competing interest:** The authors
declare that no competing
interests exist.

**Reviewing Editor:** Ping-Chih
Ho, Ludwig Institute for Cancer
Research, Switzerland

**Abstract** Leptomeningeal metastasis is associated with dismal prognosis and has few treatment options. However, very little is known about the immune response to leptomeningeal metastasis. Here, by establishing an immunocompetent mouse model of breast cancer leptomeningeal metastasis, we found that tumor-specific CD8[+] T cells were generated in deep cervical lymph nodes (dCLNs) and played an important role in controlling leptomeningeal metastasis. Mechanistically, T cells in dCLNs displayed a senescence phenotype and their recruitment was impaired in mice bearing cancer cells that preferentially colonized in leptomeningeal space. Upregulation of p53 suppressed the transcription of VLA-4 in senescent dCLN T cells and consequently inhibited their migration to the leptomeningeal compartment. Clinically, CD8[+] T cells from the cerebrospinal fluid of patients with leptomeningeal metastasis exhibited senescence and VLA-4 downregulation. Collectively, our findings demonstrated that CD8[+] T cell immunosenescence drives leptomeningeal metastasis.

## Editor's evaluation

This study address an exciting area which remains less explored. Overall, the findings presented here provides immediate impact to cancer metastasis and how the interplay between tumor cells and immune cells can affect this process.

## Introduction

Brain metastasis is associated with one of the worst clinical outcomes and has few therapeutic options (*Ahluwalia et al., 2020*). The brain contains two distinctive compartments – parenchyma mainly consisting of cells and leptomeninges filled with cerebrospinal fluid (CSF) (*Valiente et al.,*

*2018*). Many brain parenchymal cells can be found nowhere else in the body, including neurons, astrocytes, and microglia. Despite tremendous progress has been made in understanding the interaction between tumor cells and resident cells in parenchymal metastasis, very little is known about the unique microenvironment of leptomeningeal metastasis (LM) (*Boire et al., 2020*). LM, which is caused by cancer cells invading the leptomeninges, or CSF-filled spaces, represents the worst outcome of cancer patients (*Franzoi and Hortobagyi, 2019*; *Wang et al., 2018*). Despite the development of diagnostic techniques and advances in cancer treatment, the incidence of breast cancer with LM increases, ranges between 3% and 16% (*Kushner and Cheung, 1992*). LM is a fatal complication with dismal prognosis. Its median survival from detection is as low as 18 weeks, and 1-year survival is about 15% (*Niwińska et al., 2013*).

Access of circulating immune cells to the brain is limited by the blood–brain barrier (BBB). Therefore, the brain has previously been considered as an immune-privileged organ. Recently, the paradigm has shifted, as functional lymphatic vessels have been identified in meninges. Immune cells and antigens in central nervous system (CNS) can be drained to deep cervical lymph nodes (dCLNs) via meningeal lymphatic system (*Louveau et al., 2015*). Ablation of lymphatic vessels reduces the interaction between brain-specific T cells and dendritic cells (DC), and alleviates disease progression in experimental allergic encephalomyelitis (EAE) mice (*Louveau et al., 2018*), indicating that lymphatic drainage contributes to T cell activation in CNS. The very late activated Ag-4 (VLA-4) is a member of integrin family, which is widely known to mediate T cell adhesion to endothelium. In fact, VLA-4 functions as an essential adhesion molecule in T cell migration across the BBB in EAE (*Bartholomäus et al., 2009*; *Schläger et al., 2016*). However, their roles in CNS diseases are still poorly understood.

Cancer is often associated with aging as the incidences of most cancers rise dramatically in the elderly (*Bray et al., 2018*; *Aunan et al., 2017*). Oncogene activation mediates tumor cells to adopt a senescence-associated secretory phenotype (SASP), which fosters chronic inflammatory milieus (*Faget et al., 2019*; *Fane and Weeraratna, 2020*). Consequently, the aging microenvironment induces senescence of T cells, which leads to immunosuppression in multiple types of malignancies (*Ye et al., 2014*). Furthermore, the tumor microenvironment (TME) can induce senescence of adoptively transferred tumor-specific T cells, and therefore develop a resistance to immunotherapy (*Ye et al., 2013*). However, whether immunosenescence contributes to the immunosuppression of meningeal immune response to tumors remains unreported. Here, we investigated how antitumor meningeal immunity is initiated and suppressed during LM.

## Results
### Establishing a model of cancer leptomeningeal metastasis

To investigate the role of immunity in LM, we used immunocompetent mice and employed a two-stage *in vivo* selection method to enrich breast cancer cells with leptomeningeal tropism. First, selecting the cancer cells that survived in the CSF. Second, selecting cancer cells from the resulting populations for hematogenous tropism to colonize the leptomeninges through intracarotid artery injection (*Figure 1A*). Briefly, mouse breast cancer cells (EO771 and 4T1) and lung cancer cells (LLC) stably expressing luciferase were inoculated into the cisterna magna of syngeneic mice C57BL/6 (EO771-luc, LLC-luc) or BALB/c (4T1-luc) (*Figure 1A*). Once leptomeningeal metastatic lesions were established, cancer cells were isolated from the meninges and cultured *ex vivo* to obtain intermediate sublines. Intermediate cells were then inoculated into intracarotid artery for hematogenous dissemination to meningeal space. Cancer cells in metastatic lesions of leptomeningeal space after three times of enrichment were selected as a highly leptomeningeal metastatic subline "LM-phenotype cells" (*Figure 1A*). To evaluate the leptomeningeal metastatic capacity of this selected derivative, we performed bioluminescent imaging and animal MRI analysis after injection of EO771 LM-phenotype cells and their parental cells via intracarotid artery. Intense bioluminescence imaging (BLI) signals (*Figure 1B and C*) and hyperintense T1-weighted signals (*Figure 1D*) throughout tumors in the meningeal space demonstrated abundant neuro-anatomic metastases of mice injected with LM-phenotype cells, rather than the parental cells. Moreover, we confirmed the leptomeningeal localization of metastases by the immunohistochemical (*Figure 1E*, *Figure 1—figure supplement 1A*) and immunofluorescent staining (*Figure 1F*, *Figure 1—figure supplement 1B*). Furthermore, LM cell injection into the intracarotid artery significantly shortened animal survival compared to the parental cell (*Figure 1G*).

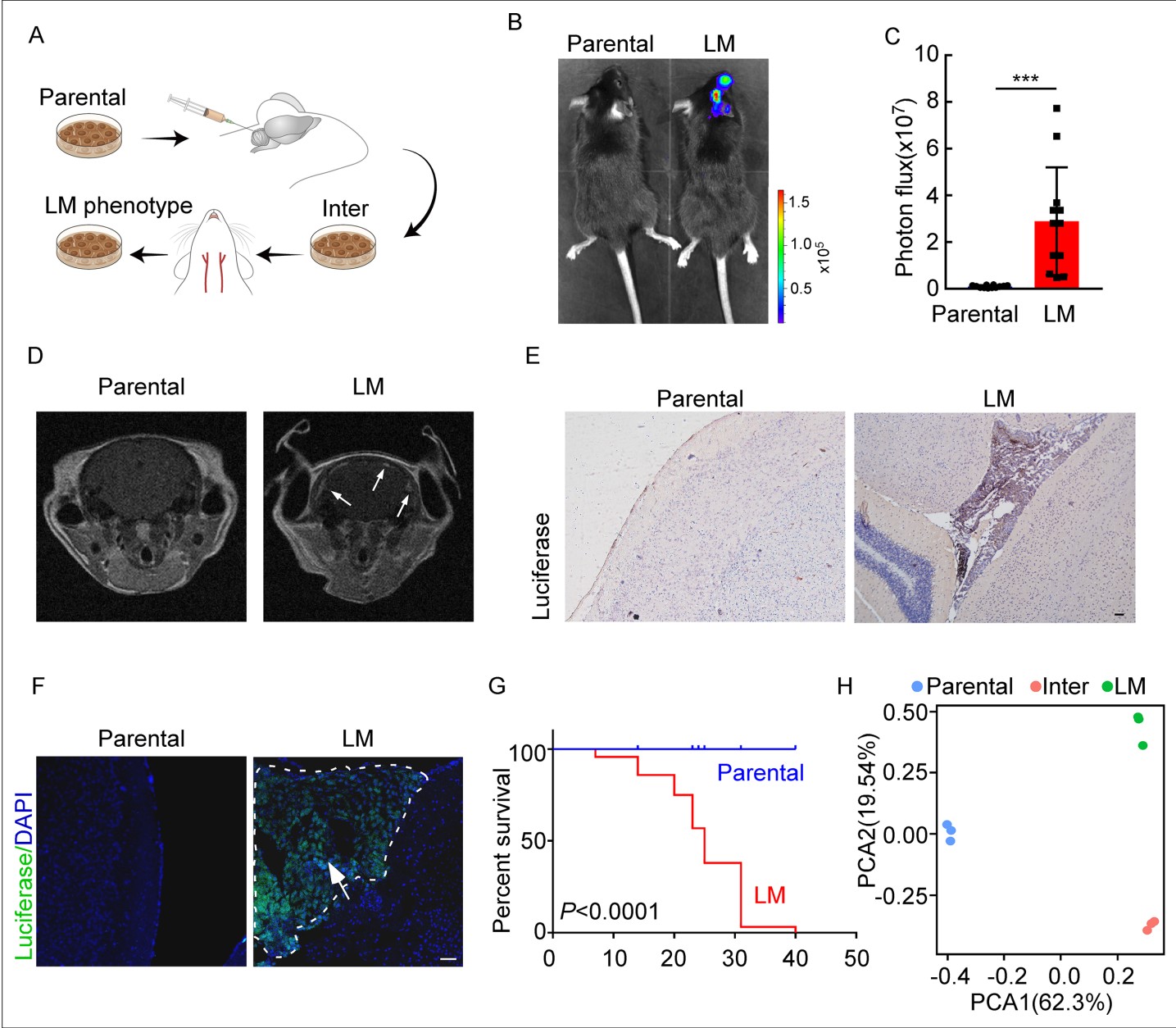

**Figure 1.** Establishing a model of breast cancer leptomeningeal metastasis (LM). (**A**) An illustration showing iterative *in vivo* selection of LM derivative cell lines. $2 \times 10^4$ tumor cells transduced with lentiviral vectors expressing luciferase were injected into the cisterna magna of recipient mice. When LM lesions were detected by IVIS, the mice were euthanized. Cells were collected and cultured before being injected into the other mice. This procedure was carried out three times to generate intermediate (Inter) cells. Next, $1 \times 10^5$ Inter cells were inoculated into the intracarotid artery. Mice bearing LM were sacrificed and tumor cells were collected from the meninges and denoted as LM derivatives. (**B–G**) $1 \times 10^5$ EO771 parental or LM-phenotype cells were inoculated into the intracarotid artery of recipient mice. (**B, C**) Tumor growth was monitored by bioluminescence imaging (BLI) imaging at day 28. Representative BLI images (**B**) and quantitation (**C**) are shown (mean ± SD, n = 12 mice per group). ***p<0.001 by two-tailed Student's *t* test. (**D**) MRI at day 28 post-inoculation revealed LM was formed after LM-phenotype cell inoculation. The white arrows indicate metastatic lesions. (**E**) Neuro-anatomic localization of metastases was determined by immunohistochemical staining. Scale bar = 50 µm. (**F**) Representative pictures of immunofluorescent staining for LM lesions in mice (luciferase, green; DAPI, blue). The white line indicates the border of the metastatic lesions. Scale bar = 50 µm. (**G**) Kaplan–Meier plot of overall survival of mice. (**H**) Principal component analysis (PCA) plots of gene expression data showing a segregation among parental (blue), LM-phenotype (green), and inter (orange) cell lines. Genes with base mean ≥ 50, fold change ≥2 or ≤0.5 and p<0.01 were included for analysis.

The online version of this article includes the following source data and figure supplement(s) for figure 1:

**Source data 1.** RNA sequencing data generated in *Figure 1H*.

**Figure supplement 1.** Establishing a model of leptomeningeal metastasis (LM).

Similar results were observed in mice injected with 4T1 LM-phenotype cells (*Figure 1—figure supplement 1C–F*) and LLC LM-phenotype cells (*Figure 1—figure supplement 1G–J*). Furthermore, principal component analysis (PCA) of the transcriptome confirmed that the gene expression profiles of EO771 parental, inter and LM-phenotype cells segregate independently (*Figure 1H*). Thus, there was a heterogeneity between LM population and its matched parental population.

## CD8+ T cells constrain leptomeningeal metastasis

Emerging data show that meningeal immunity plays a crucial role in various diseases (*Ransohoff and Engelhardt, 2012*), but how this CNS barrier operates immunologically under LM remains poorly understood. We found that CD45+ immune cells increased in mice injected with parental or LM cells, compared with those injected with PBS, indicating that the meninges were inflamed after tumor inoculation. However, meningeal immune cell numbers in the presence of LM dropped significantly in comparison with those injected with parental cells (*Figure 2A*). To further investigate the changes of meningeal immune repertoire during LM, we analyzed the absolute numbers of meninge-infiltrating immune cells, including T cells, monocytes, microglia, myeloid cells, and neutrophils (*Manglani et al., 2018*). Interestingly, we observed that only T cells were markedly decreased in LM (*Figure 2B*, *Figure 2—figure supplement 1A*). To confirm the role of T lymphocytes in LM, we injected EO771 LM-phenotype cells into *Rag2-/-* mice, which carry a targeted knockout mutation in recombination activating gene 2 (*Rag2*) and therefore lack mature T or B cells (*Fan et al., 2019*). Compared with wild type (WT) C57BL/6 mice, accelerated intracranial tumor progression was observed in the *Rag2-/-* mice when LM-phenotype cells were injected via intracarotid arteries, which was evaluated by bioluminescent signals (*Figure 2C and D*). Similarly, we injected 4T1 LM-phenotype cells into the immunodeficient nude mice and immunocompetent BALB/c mice, and observed that tumor formation was significantly enhanced in nude mice (*Figure 2—figure supplement 1B and C*). To assess which subset of T cells contributes to constraining LM, we depleted CD4+ and CD8+ T cells in immunocompetent C57BL/6 mice via anti-CD4 (α-CD4) and anti-CD8 neutralizing antibodies (α-CD8), respectively. Analyzing the LM lesions by histopathology, we found that intracranial tumor growth dramatically increased in the mice with CD8+ T cell depletion compared with IgG neutralization (*Figure 2E*, *Figure 2—figure supplement 1D*). By contrast, depletion of CD4+ T cells even exhibited slight tumor regression, although the difference did not reach statistical significance (*Figure 2E*, *Figure 2—figure supplement 1D*). These results suggested that CD8+ T cells play an important role in constraining intracranial tumor growth.

Most of the immune surveillance takes place in the tumor draining lymph nodes (TDLN) largely as a result of its unique cellular composition and proximity to the primary tumor (*Chandrasekaran and King, 2014*). It has been reported that dCLNs communicate with meningeal lymphatics directly (*Louveau et al., 2015*). Evans blue, which is widely used as a marker of the BBB integrity, can be preferentially drained via the lymphatics (*Maloveska et al., 2018*). Therefore, to visualize CNS lymphatic drainage, we injected Evans blue into the cisterna magna and detected the presence of dye in dCLNs 30 min later (*Figure 2F*). Furthermore, we isolated CD4+ and CD8+ T cells from dCLNs of WT mice 2 weeks after EO771 cell injection. These isolated T cells were adoptively transferred into syngeneic *Rag2-/-* mice, respectively, followed by LM-phenotype cell inoculation (*Figure 2G*). Interestingly, intracranial tumors arising in *Rag2-/-* mice was much smaller when receiving transferred CD8+ T cells, but not CD4+ T cells (*Figure 2H*, *Figure 2—figure supplement 1E*). More importantly, mice receiving CD8+ T cells had longer overall survival than those receiving CD4+ T cells or PBS (*Figure 2I*). These data suggested that CD8+ T cells are crucial effectors in controlling LM.

## dCLNs generate tumor-specific CD8+ T cells against leptomeningeal metastasis

The priming of tumor-specific CD8+ T cells primarily relies on antigen-presenting DCs, which is the fundamental step that launches T cell response against tumor (*Bandola-Simon and Roche, 2019*; *Alfei et al., 2021*). To explore whether tumor-specific CD8+ T cells were primed by DCs in dCLN, we firstly examined the antigen processing of DCs. We inoculated EO771 breast tumor cells with ectopic expression of chicken ovalbumin (OVA) (EO771-OVA) into the cisterna magna of C57BL/6 mice and isolated the CD11c+ DCs from the draining dCLNs, non-draining inguinal LNs, and spleen 7 days later. Interestingly, OVA peptide/MHC class I complex, SIINFEKL/H2-Kb, was only detected on CD11c+ DCs

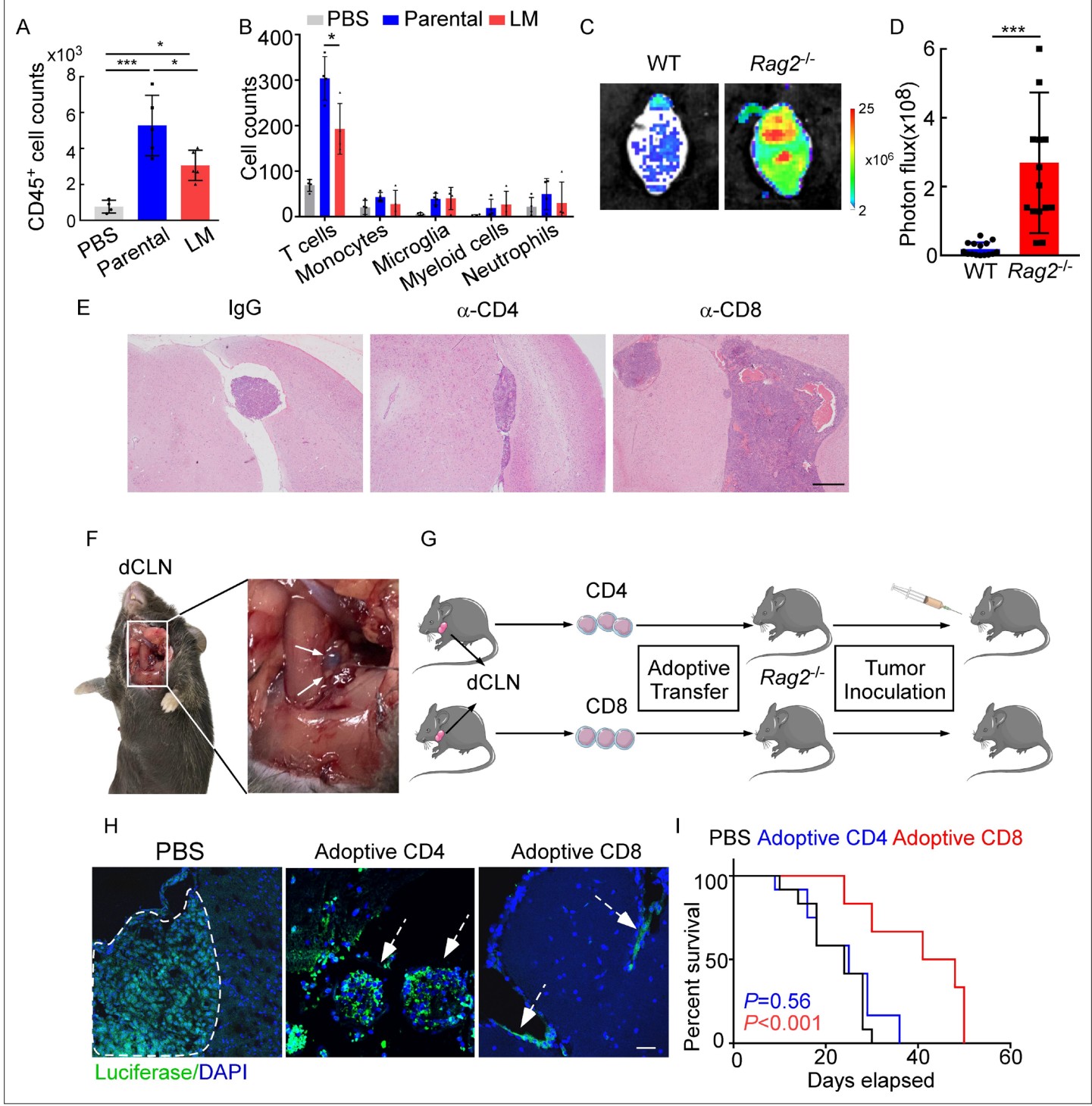

**Figure 2.** CD8+ T cells constrain leptomeningeal metastasis (LM). (**A–B**) PBS, $1 \times 10^5$ EO771 parental or LM-phenotype cells were inoculated into the intracarotid artery of recipient mice. (**A**) Histogram represents the absolute number of meningeal CD45+ immune cells (mean ± SD. PBS n = 4 per group; parental, LM n = 5 per group). *p<0.05, ***p<0.001 by one-way ANOVA with Tukey's multiple-comparison test. (**B**) Histogram represents the absolute number of diverse meningeal immune cells mice (mean ± SD, n = 4 per group). *p<0.05 by one-way ANOVA with Tukey's multiple-comparison test. (**C, D**) $1 \times 10^5$ EO771 LM-phenotype cells were injected into the intracarotid artery of wild type C57BL/6 (WT) or $Rag2^{-/-}$ mice. Representative images (**C**) and quantitation (**D**) for tumor growth monitored by bioluminescence imaging (BLI) at day 28 post-injection (mean ± SD, n = 12 per group.). ***p<0.001 by two-tailed Student's *t* test. (**E**) $1 \times 10^5$ LM-phenotype cells were inoculated into the intracarotid artery of WT C57BL/6 mice treated with IgG, anti-CD4 (α-CD4), or anti-CD8 neutralizing antibodies (α-CD8), respectively. Representative images for intracranial tumor lesions are shown. Scale bar = 20 μm. (**F**) Evans blue was injected into the cisterna magna of WT C57BL/6 mice. The presence of dye in deep cervical lymph nodes (dCLNs) was detected after

*Figure 2 continued on next page*

Figure 2 continued

30 min. Representative images of the Evans blue accumulation in the dCLN. The white arrowhead points to the dCLN. (**G–I**) CD4+ and CD8+ T cells were isolated from dCLNs of WT mice injected with EO771 cells and adoptively transferred into *Rag2-/-* mice, followed by the inoculation of LM-phenotype cells. (**G**) Schematics of the adoptive cell transfer model in immunocompetent mice. (**H**) Representative immunofluorescence staining for LM lesions in mice with indicated treatment (luciferase, green; DAPI, blue). The white line and arrows indicate the border of the metastatic lesions. Scale bar = 50 µm. (**I**) Kaplan–Meier plots of overall survival of mice with indicated treatment. Black, PBS (n = 12); blue, adoptive transfer of CD4+ T cells (n = 12, p=0.56 compared with mice treated with PBS); red, adoptive transfer of CD8+ T cells (n = 12, p<0.001 compared with mice treated with PBS).

The online version of this article includes the following figure supplement(s) for figure 2:

**Figure supplement 1.** CD8+ T cells play a role in constraining intracranial tumor growth.

in dCLNs but not the ones in inguinal LNs and spleen (*Figure 3A*, *Figure 3—figure supplement 1A and B*). Furthermore, we employed H-2-Kb-OVA/SIINFEKL pentamer staining to detect the generation of OVA-specific CD8+ T cells. Interestingly, we observed that a significant proportion of OVA/SIINFEKL pentamer+ CD8+ T cells was detected in the dCLNs of mice injected with EO771-OVA cells, but not in the mice injected with PBS, wild type EO771 (EO771 WT) and EO771 with an irrelevant antigen glycoprotein B (EO771-gB) cells (*Figure 3B*, *Figure 3—figure supplement 1C*). Moreover, CD11c+ DCs were subsequently isolated from dCLNs, and then co-cultured with naïve CD8+ T cells isolated from the spleens of OT-1 mice in vitro (*Figure 3C*). Analysis of OT-1 T cell priming revealed that only DCs from mice receiving EO771-OVA inoculation could expand OT-1 T cells (*Figure 3D*, *Figure 3—figure supplement 1D*) and induce a significant increase in IFN-γ production (*Figure 3E*, *Figure 3—figure supplement 1E*) of OT-1 T cells.

Meningeal lymphatics act as an avenue for CNS drainage and immune cell trafficking (*Tavares and Louveau, 2021*). We performed the surgical ligation of the lymphatic afferent to the dCLNs, which could disrupt the meningeal lymphatic drainage. A week after the surgery, sham or ligation group was inoculated with EO771-OVA cells, respectively (*Figure 3F*). Determined by the expression of SIINFEKL/H2-Kb complex on the surface of CD11c+ dCLN DCs, antigen processing was impaired in the mice receiving surgical ligation (*Figure 3G*, *Figure 3—figure supplement 1F*). In addition, the proliferation (*Figure 3H*, *Figure 3—figure supplement 1G*) and IFN-γ production (*Figure 3I*, *Figure 3—figure supplement 1H*) of OT-1 T cells primed by dCLN DCs were diminished. Collectively, dCLNs could generate tumor-specific CD8+ T cells.

## dCLN CD8+ T cells exhibit senescence in leptomeningeal metastasis

Adaptive immune response against tumor was provoked by antigen-specific CD8+ T cells (*Kim et al., 2007*). Whether CD8+ T cells control metastasis in metastatic locations or affecting dissemination of cancer cells is not clear. We first evaluated whether CD8+ T cells affect the dissemination of cancer cells by analyzing disseminated tumor cells (DTCs) in peripheral blood of C57BL/6 mice injected with EO771-luc parental or LM cells. The DTCs in the blood were defined as CD45-luciferase+ cells by flow cytometry (*Figure 4—figure supplement 1A*). We found that the percentages of DTCs were not significantly different between these two groups (*Figure 4—figure supplement 1B and C*). Moreover, we injected EO771-luc LM cells into *Rag2-/-* mice and transferred CD8+ T cells later. We found that CD8+ T cell transfusion did not influence the percentages of DTCs in the peripheral blood (*Figure 4—figure supplement 1D*). These data suggested that CD8+ T cells did not affect the dissemination of cancer cells, probably the metastatic lesion. Then we observed that in the meninge, the number of CD8+ T cells of mice injected with EO771 LM-phenotype cells (LM-CD8+ T cells) was much lower than the one in mice injected with parental cells (Parental-CD8+ T cells) (*Figure 4A and B*, *Figure 4—figure supplement 1E*). Then, we further investigated the reason for the drop of CD8+ T cell count in the meninges. It is well known that tumor-specific CD8+ T cells were primed by DCs in dCLNs, migrated to meninges, and recognized and lysed tumor cells (*Calzascia et al., 2005*). Therefore, we evaluated the number of CD8+ T cells in dCLNs of mice injected with LM cells and parental cells. Interestingly, we found that the absolute count of LM-CD8+ T cells in dCLNs was significantly lower than Parental-CD8+ T cells (*Figure 4C and D*, *Figure 4—figure supplement 1F*). Moreover, the proliferation of LM-CD8+ T cells in dCLNs was significantly decreased, compared with the dCLN Parental-CD8+ T cells, determined by flow cytometric analysis of CD8 and Ki-67 co-staining (*Figure 4E*, *Figure 4—figure supplement 1G*). These data suggested that the decreased proliferation of CD8+ T cells in dCLNs was one of the reasons of low infiltration of CD8+ T cells in LM lesion.

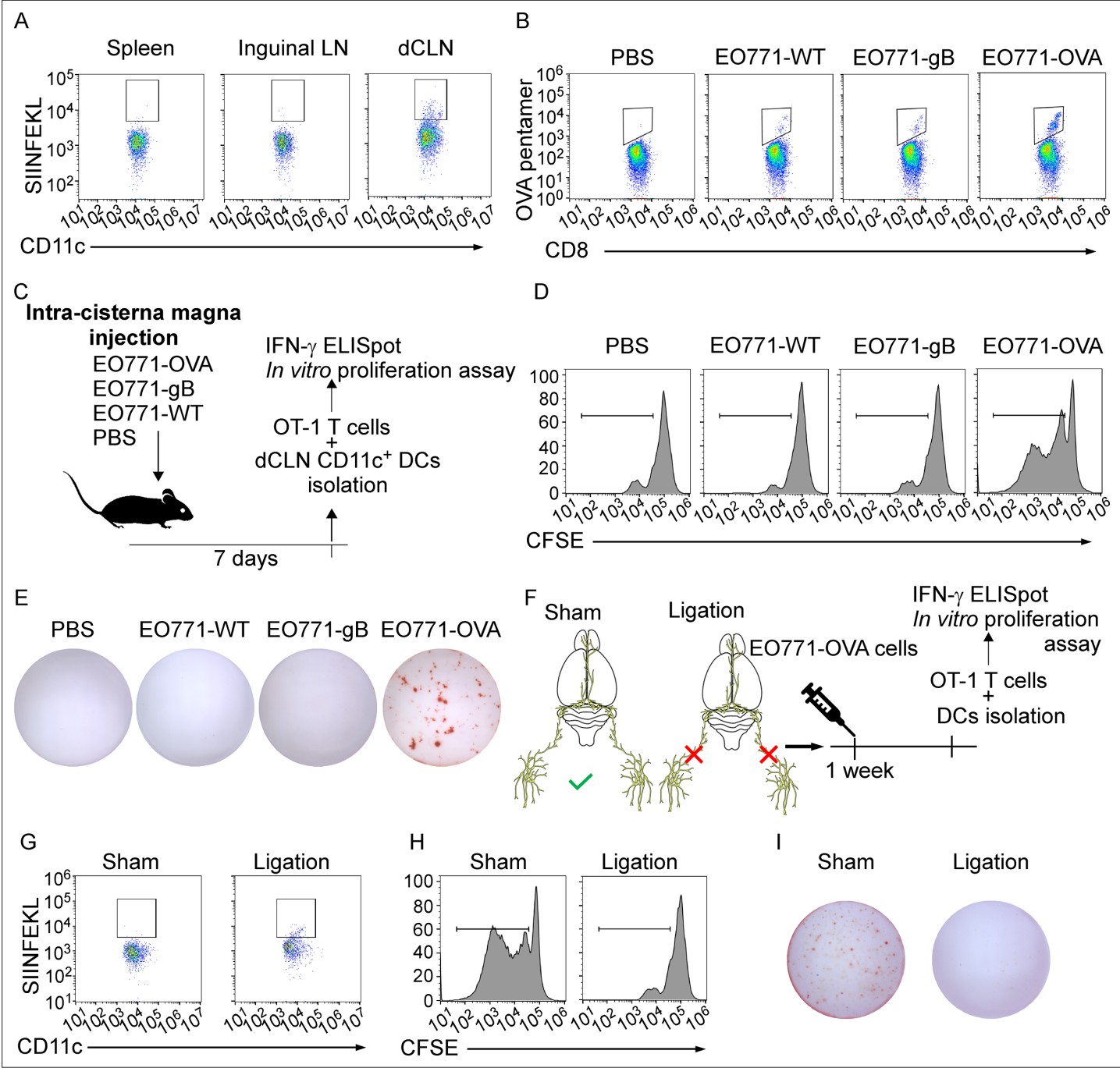

**Figure 3.** Deep cervical lymph nodes (dCLNs) generate tumor-specific CD8+ T cells against leptomeningeal metastasis (LM). (**A**) Dendritic cells (DCs) from spleen, inguinal lymph nodes (LNs), and dCLNs were isolated 7 days after intra cisterna magna EO771-OVA cell injection, and later analyzed for SIINFEKL presentation (n = 8 per group). (**B**) PBS, EO771, and EO771 expressed with gB (EO771-gB) and OVA (EO771-OVA) were injected into C57BL/6 mice. CD8+ T cells from LNs were isolated and later analyzed for ovalbumin (OVA) pentamer expression. (**C**) Schematics of animal experiments detecting the generation of tumor-specific CD8+ T cells in vitro. PBS, EO771, and EO771-gB and EO771-OVA were injected into C57BL/6 mice. DCs were subsequently isolated from dCLNs, and then co-cultured with OT-1 T cells in vitro. (**D**) Representative histogram of CFSE dilution of OT-1 T cells co-cultured with CD11c+ cells isolated from dCLNs of mice with indicated treatment for 60 hr. (**E**) Representative images of IFN-γ ELISpot data of OT-1 T cells co-cultured with CD11c+ cells isolated from dCLNs of mice with indicated treatment for 60 hr. (**F**) Schematics of animal experiments illustrating dCLNs generate tumor-specific CD8+ T cells against LM. Surgical ligation of the lymphatics afferent to the dCLNs was performed. A week after the surgery, sham or ligation group was inoculated with EO771-OVA cells. DCs were subsequently isolated from dCLNs, and then co-cultured with OT-1 T cells in vitro. (**G**) Representative histogram of SIINFEKL expression on CD11c+ cells isolated from dCLNs of mice with indicated treatment. (**H**) Representative histogram of CFSE dilution of OT-1 T cells co-cultured with CD11c+ cells isolated from dCLNs of mice with indicated treatment

*Figure 3 continued on next page*

*Figure 3 continued*

for 60 hr. (**I**) Representative images of IFN-γ ELISpot data of OT-1 T cells co-cultured with CD11c⁺ cells isolated from dCLNs of mice with indicated treatment for 60 hr.

The online version of this article includes the following figure supplement(s) for figure 3:

**Figure supplement 1.** Deep cervical lymph nodes (dCLNs) generate antigen-specific CD8⁺ T cells against leptomeningeal metastasis (LM).

Since T cell proliferation is often impaired during apoptosis and senescence, we evaluated the apoptosis and senescence of LM- and Parental-CD8⁺ T cells in dCLNs. Determined by the apoptotic marker, cleaved caspase 3 expression, by flow cytometric analysis, LM- and Parental-CD8⁺ T cells have equivalent amount of cleaved caspase 3 (*Figure 4F*, *Figure 4—figure supplement 1H*), indicating that apoptosis is not the major cause contributing to the impaired proliferation of T cells. On the other hand, senescent cells exhibit stress signs, which is characterized by withdrawal from the cell cycle and loss of the capability to proliferate under stimuli of growth factors or mitogens (*Montes et al., 2008*). We detected the distinct increased expression of senescence markers p53 and p21 in dCLN LM- CD8⁺ T cells compared with dCLN Parental-CD8⁺ T cells (*Figure 4G*, *Figure 4—source data 1*). Moreover, the percentage of senescence-associated β-galactosidase (SA-β-gal)-positive T cells was higher in the LM-CD8⁺ T cells (*Figure 4H and I*). Similar results were also found in CD8⁺ T cells in dCLNs of the mice injected with LLC LM-phenotype cells (*Figure 4—figure supplement 1I–K*). Collectively, CD8⁺ T cells in dCLNs of mice bearing LM exhibit senescence.

## Downregulation of VLA-4 in senescent CD8⁺ T cells impairs their trafficking to meninges

Besides the proliferation, we also evaluated the trafficking ability of dCLN LM- and Parental- CD8⁺ T cells. We isolated the dCLN CD8⁺ T cells, labeled with CFSE (Carboxyfluorescein succinimidyl ester) and injected into the tail vein of recipient mice. Two-photon live imaging showed that the migration of LM- CD8⁺ T cells to meninges *in vivo* was significantly less than Parental-CD8⁺ T cells (*Figure 5A*, *Figure 5—figure supplement 1A*), suggesting that CD8⁺ T cells displayed impaired trafficking ability to meninges under LM.

The very late activated Ag-4 (VLA-4)-vascular adhesion molecule-1 (VCAM-1) pathway plays an important role in T cell leptomeningeal recruitment (*Theien et al., 2003*). Given the reduction in CD8⁺ T cell trafficking to meninges, we evaluated VLA-4 levels in meningeal and dCLN CD8⁺ T cells by flow cytometric analysis. We observed that VLA-4 levels in CD8⁺ T cells from both leptomeningeal space and those from dCLNs were downregulated in mice injected with LM-phenotype cells, compared with those of mice injected with parental cells (*Figure 5B*, *Figure 5—figure supplement 1B*). Similar results were found in BALB/c mice injected with 4T1 parental and LM cells (*Figure 5—figure supplement 1C*). To evaluate the function of VLA-4 in breast cancer LM, we applied neutralization antibody against VLA-4 and found that VLA-4 blockade decreased the number of CD8⁺ T cell in meninges (*Figure 5C*), and aggregated intracranial tumor metastasis, determined by histopathology (*Figure 5D*) and BLI signaling (*Figure 5E and F*).

To further investigate the role of VLA-4 in T cell trafficking, we have examined the contribution of VLA-4 in the adhesion and migration of CD8⁺ T cells. LM- or Parental-CD8⁺ T cells were isolated from dCLNs and tested for their ability to adhere to plate-bound VCAM-1 protein (*Figure 5G*; *Boire et al., 2020*). Parental-CD8⁺ T cells with higher expression of VLA-4 exhibited specific adhesion to plate-bound VCAM-1 protein (74.77 ± 4.27%), whereas LM- CD8⁺ T cells with lower expression of VLA-4 displayed only background levels of adhesion (9.96 ± 0.89%). Pretreatment of Parental-CD8⁺ T cells with the anti-VLA-4 neutralizing antibody virtually ablated their ability to adhere to VCAM-1 (19.22 ± 3.44%), while LM-CD8⁺ T cells pretreated with the anti-VLA-4 antibody showed no difference in their ability to adhere to VCAM-1 (8.43 ± 0.85%) (*Figure 5G*). To investigate the role of VLA-4 in T cell migration, we employed an in vitro BBB model to assess the transmigration of T cells. CD8⁺ T cells were added to the top chamber and treated with IgG or VLA4 antibody. We observed that the migration of Parental-CD8⁺ T cells was much stronger than LM-CD8⁺ T cells, which was disrupted by anti-VLA4 treatment (*Figure 5H*). Furthermore, VLA-4 blockade had no effect on the proliferation (*Figure 5I*, *Figure 5—figure supplement 1D*) and cell death (*Figure 5J*, *Figure 5—figure supplement 1E*) of dCLN CD8⁺ T cells. Taken together, these data suggested that blocking VLA-4 in CD8⁺

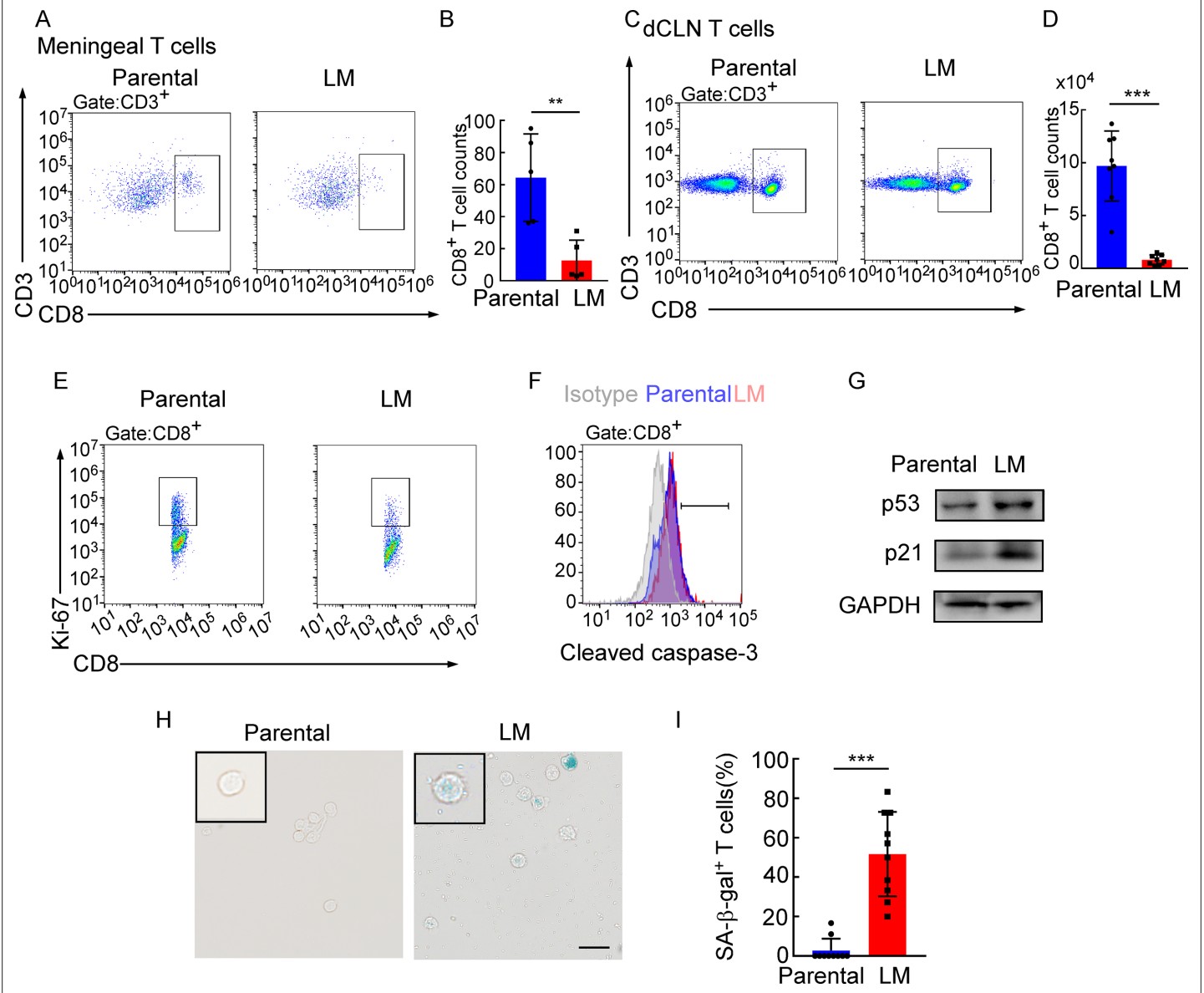

**Figure 4.** Deep cervical lymph node (dCLN) CD8+ T cells exhibit senescence in leptomeningeal metastasis (LM). (**A–I**) 1 × 10⁵ EO771 parental or LM-phenotype cells were inoculated into the intracarotid artery of recipient mice. (**A, B**) T cells in the meninges were isolated from mice injected with EO771 parental and LM-phenotype cells and analyzed by flow cytometry. Representative images (**A**) and quantitation (**B**) of meningeal CD8+ T cells in gated CD3+ T cells are shown (mean ± SD, n = 5 per group). **p<0.01 by two-tailed Student's *t* test. (**C, D**) T cells in the dCLNs were isolated from mice injected with EO771 parental and LM-phenotype cells and analyzed by flow cytometry. Representative images (**C**) and quantitation (**D**) of dCLN CD8+ T cells in gated CD3+ T cells are shown (mean ± SD, n = 8 per group). ***p<0.001 by two-tailed Student's *t* test. (**E**) Representative images of proliferative capacity of CD8+ T cells isolated from dCLNs, as determined by flow cytometry for the percentages of Ki-67+ cells (n = 4 per group). (**F**) Representative histogram of apoptosis of CD8+ T cells isolated from dCLNs, as determined by flow cytometry for the percentages of cleaved caspase-3+ cells. Gray, isotype; blue, parental; red, LM. (**G**) Representative immunoblots for p53 and p21 in CD8+ T cells isolated from dCLNs. (**H, I**) Representative images (**H**) and quantitation (**I**) of SA-β-gal staining in CD8+ T cells isolated from dCLNs. Scale bar = 20 μm (mean ± SD, n = 10 per group). ***p<0.001 by two-tailed Student's *t* test.

The online version of this article includes the following source data and figure supplement(s) for figure 4:

**Source data 1.** Raw data of Western Blots in *Figure 4G*.

**Figure supplement 1.** Meningeal CD8+ T cells that show cell cycle arrest undergo senescence instead of apoptosis under leptomeningeal metastasis (LM).

**Figure supplement 1—source data 1.** Raw data of Western Blots in *Figure 4—figure supplement 1I*.

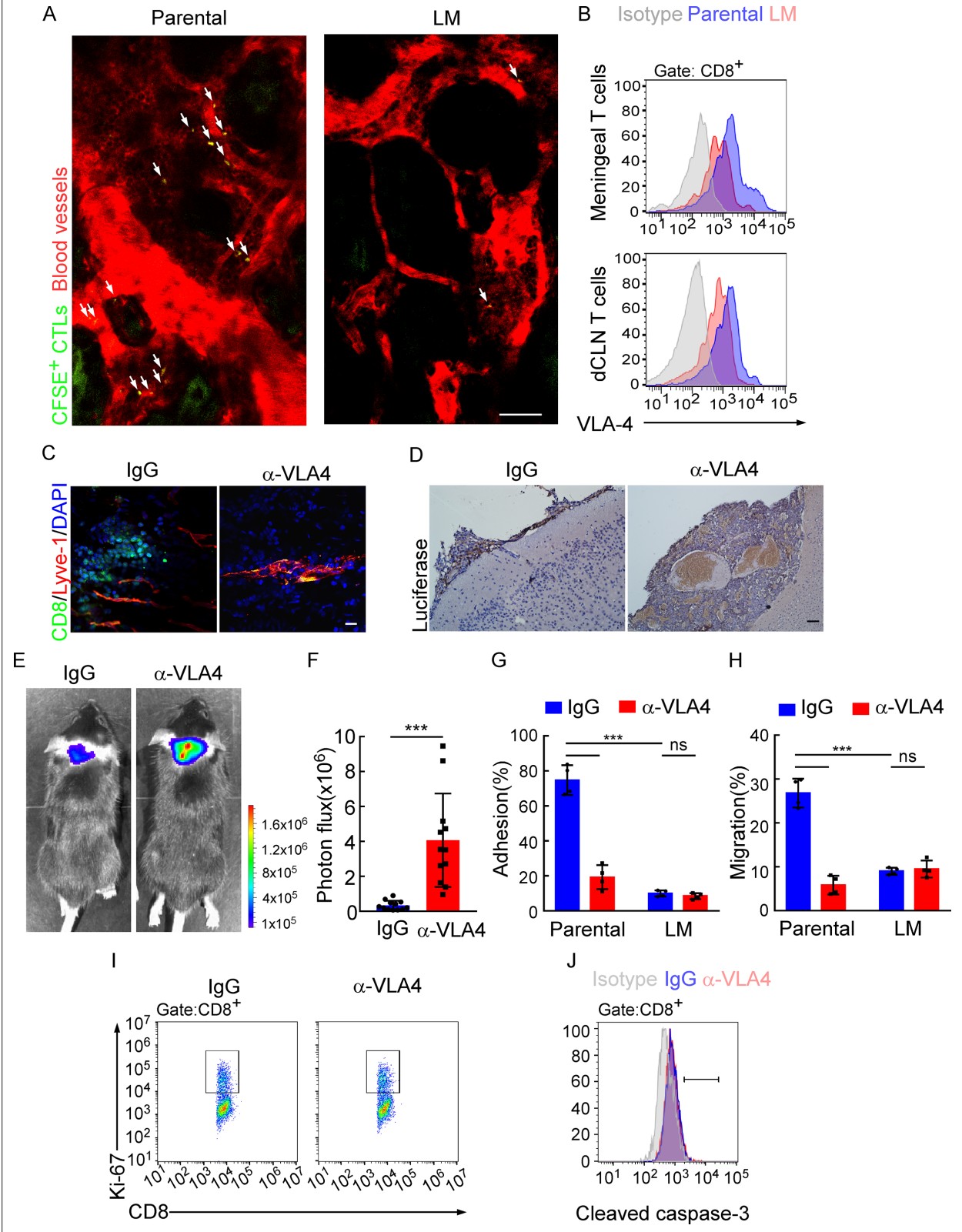

**Figure 5.** Downregulation of VLA-4 in senescent CD8[+] T cells impairs their trafficking to meninges. (**A**) CD8[+] T cells were isolated from the deep cervical lymph nodes (dCLNs) of mice injected with parental and leptomeningeal metastasis (LM)-phenotype cells, respectively, stained with CFSE, and subsequently transferred to the recipient mice. 24 hr after T cell transfusion, two-photon imaging was used to reveal the *in vivo* migration of CFSE-labeled CTLs to meninges. Visualization of the vasculature by i.v. injection of rhodamine-dextran (red). The location of CFSE[+] T cells (green) is marked

*Figure 5 continued on next page*

*Figure 5 continued*

by a white arrowhead. Scale bar, 50 µm. (**B**) T cells in the meninges and dCLNs were isolated from mice injected with parental and LM-phenotype cells. Flow cytometry analysis of VLA-4 expression in CD8$^+$ T cells isolated from meninges (top) or dCLNs (bottom) of mice receiving parental or LM-phenotype cells. Gray, isotype; blue, Parental-CD8$^+$ T cells; red, LM-CD8$^+$ T cells. (**C–F**) C57BL/6 mice pretreated with IgG or α-VLA-4 antibodies were injected with parental EO771-luc cells via intracarotid artery. (**C**) Representative immunofluorescent images of meningeal CD8$^+$ T cells from C57BL/6 mice with indicated treatment in whole-mount meninges. Scale bar = 50 µm. Red, Lyve-1; green, CD8; blue, DAPI. (**D**) Representative IHC images for luciferase to identify LM lesions. Scale bar = 50 µm. (**E, F**) Representative bioluminescence images (**E**) and quantitation (**F**) of metastases in mice with indicated treatment at day 21 post-injection (mean ± SD, n = 12 per group). \*\*\*p<0.001 by two-tailed Student's *t* test. (**G**) CD8$^+$ T cells in dCLNs of mice injected with EO771 LM-phenotype cells or parental cells were isolated and tested for their ability to adhere to plate-bound VCAM-1-Ig fusion protein. Histogram shows the number of cells adherent to the bottom of the wells under indicated treatments (mean ± SD, n = 4 per group). \*\*\*p<0.001, ns, not significant by two-way ANOVA with Sidak's multiple-comparison test. (**H**) CD8$^+$ T cells isolated from mice injected with EO771 LM-phenotype cells or parental cells were treated with IgG or VLA4 neutralizing antibody and later added to the top chamber of in vitro blood–brain barrier model. Histogram indicates the number of migrated CD8$^+$ T cells after IgG or VLA4 antibody treatment(mean ± SD, n = 4 per group). \*\*\*p<0.001, ns, not significant by two-way ANOVA with Sidak's multiple-comparison test. (**I**) Representative images of proliferative capacity of dCLN CD8$^+$ T cells under indicated treatment, as determined by flow cytometry for the percentages of Ki-67$^+$ cells. (**J**) Representative histogram of apoptosis of dCLN CD8$^+$ T cells under indicated treatment, as determined by flow cytometry for the percentages of cleaved caspase-3$^+$ cells. Gray, isotype; blue, IgG antibody; red, VLA-4 antibody.

The online version of this article includes the following figure supplement(s) for figure 5:

**Figure supplement 1.** Downregulated VLA-4 in CTLs inhibits their recruitment to meninges and their capacity to control leptomeningeal metastasis (LM).

---

T cells inhibited their adhesion and migration, leading to the impaired recruitment to meninges and lack of the capacity to control LM.

## VLA-4 transcription is repressed by p53

To further explore the mechanisms for VLA-4 downregulation in LM-CD8$^+$ T cells, we predicted the possible transcription factors binding to VLA-4 promoter by three different algorithms (JASPAR, PROMO, and TFBIND). Interestingly, we found p53, the senescence marker, which functions as a transcription factor involved in a myriad of cellular activities (*Wang et al., 2009*). To explore whether VLA-4 transcription is controlled by p53 signaling, we examined the VLA-4 expression in the CD8$^+$ T cells isolated from dCLNs of WT and p53-deficient (*Trp53$^{-/-}$*) mice injected with LM-phenotype cells. We found that VLA-4 levels in dCLN CD8$^+$ T cells were lower in WT mice injected with LM-EO771 cells. By comparison, p53 knockout restored VLA-4 levels of CD8$^+$ T cells in *Trp53$^{-/-}$* mice inoculated with LM-EO771 cells (*Figure 6A*), suggesting that p53 mediated VLA-4 suppression in CD8$^+$ T cells. In silico analyses performed with three different algorithms (JASPAR, PROMO, and TFBIND) predicted one putative p53-binding site located at –1550 to –1525 bp upstream of the VLA-4(*Itga4*) transcription start site (TSS) (*Figure 6B*). We electronically transfected pGL3 reporter plasmids containing the WT or mutant luciferase constructs of p53-binding sites of 5'-flanking region upstream of *Itga4* into EL4 lymphoma cells, a T lymphoma cell line previously used for gene transcription studies (*Kao et al., 2011*). Upon p53 forced expression, the luciferase signal of EL4 cells transfected with full-length *Itga4* TSS decreased. Moreover, mutation of p53-binding site abolished the suppression of luciferase activity (*Figure 6C*). To confirm these results in an endogenous setting, we performed chromatin immunoprecipitation (ChIP)–qPCR with an antibody against p53 and found an average of 4.26-fold enrichment was obtained with anti-p53 antibody in the LM-CD8$^+$ T cells compared to ChIP with a control immunoglobulin G (IgG). By contrast, an average of 1.03-fold enrichment was found in Parental-CD8$^+$ T cells (*Figure 6D*). To further illustrate the contribution of p53 in LM, we injected LM-phenotype or parental tumor cells in WT and *Trp53$^{-/-}$* mice. In *Trp53$^{-/-}$* mice, the senescence of CD8$^+$ T cells was prevented as a result of p53 deficiency (*Figure 6—figure supplement 1A and B*), contributing to upregulation of VLA-4 (*Figure 6A*) and enhanced trafficking ability to meninges (*Figure 6—figure supplement 1C and D*). Therefore, tumor growth was inhibited in P53-deficient mice determined by BLI signal (*Figure 6—figure supplement 1E and F*). Taken together, these results demonstrated that VLA-4 transcription in T cells in LM is suppressed by the senescence factor p53.

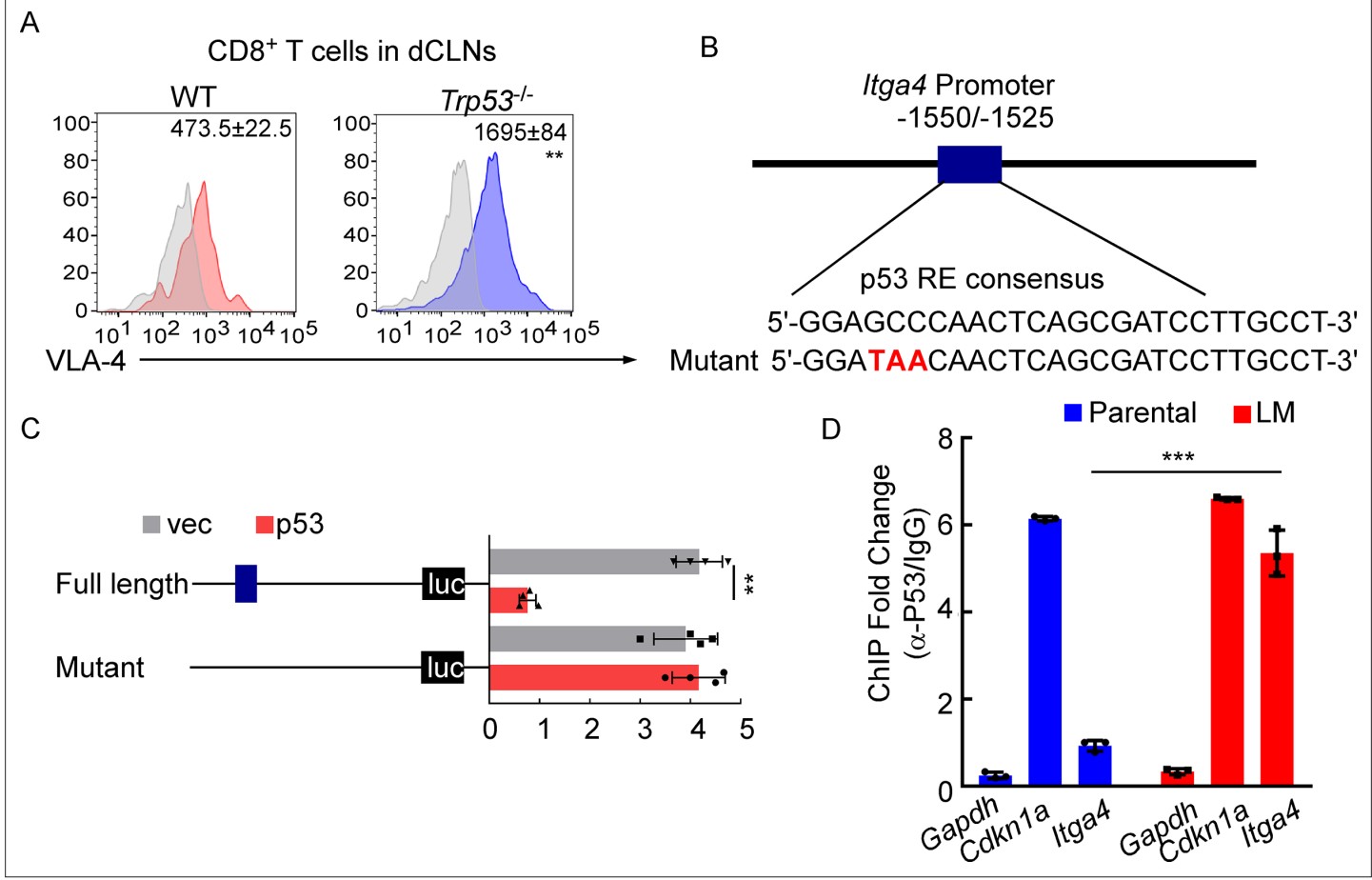

**Figure 6.** VLA-4 transcription is repressed by p53. (**A**) CD8[+] T cells were isolated from deep cervical lymph nodes (dCLNs) of wild type (WT) or *Trp53[-/-]* mice injected with leptomeningeal metastasis (LM)-phenotype cells. Flow cytometry analysis of VLA-4 expression in CD8[+] T cells from dCLNs in WT mice or *Trp53[-/-]* mice receiving LM-phenotype cells. Gray, isotype; blue, CD8[+] T cells from WT mice; red, CD8[+] T cells from *Trp53[-/-]* mice. Numbers in plot correspond to the mean fluorescent intensity of VLA-4 in CD8[+] T cells (mean ± SD, n = 4 per group). **p<0.01 compared with CD8[+] T cells from WT mice by two-tailed Student's *t* test. (**B**) A schematic of *Itga4* gene promoter. P53 binding site is identified. (**C**) EL4 cells were transfected with WT or a mutant version in which the putative p53 binding site was mutated with four nucleotides (mutant). Afterward, EL4 cells were transfected with empty vector or P53 overexpression plasmids and harvested for the luciferase activity assay (mean ± SD, n = 4 biologically independent experiments). Results are expressed relatively to control conditions. **p≤0.01 by one-way ANOVA with Tukey's multiple-comparison test. (**D**) ChIP was performed with a p53-targeting antibody or a control IgG to assess p53 binding to the *Itga4* promoter in CD8[+] T cells isolated from mice injected with parental or LM-phenotype cells (mean ± SD, n = 3). *Cdkn1a* serves as a positive control, *Gapdh* as a negative control for p53 binding. ***p<0.001 by two-tailed Student's *t* test.

The online version of this article includes the following figure supplement(s) for figure 6:

**Figure supplement 1.** The role of P53 in leptomeningeal metastasis (LM).

## Meningeal CD8[+] T cells exhibit VLA-4 downregulation and senescence in human leptomeningeal metastasis

To evaluate whether this finding in the animal model is consistent with patients with LM, we investigated VLA-4 expression and the senescence phenotype in human T cells isolated from CSF of 145 patients with nonmalignant neurological diseases and 45 patients with leptomeningeal involvements, including 6 cases of breast cancer, 35 cases of lung cancer, and 4 cases of gastrointestinal cancer (***Figure 7A***). We found that the proportion of SA-β-gal[+]CD8[+] T cells was higher in CSF from metastatic patients compared with those of nonmalignant neurological disease patients (***Figure 7B***, ***Figure 7—figure supplement 1A***), indicating that intracranial CD8[+] T cells experienced senescence in patients with LM. Moreover, flow cytometry analysis showed the CSF CD8[+] T cells of metastatic patients had lower expression of VLA-4 (***Figure 7C***). Moreover, the percentage of SA-β-gal[+]CD8[+] T cells was negatively

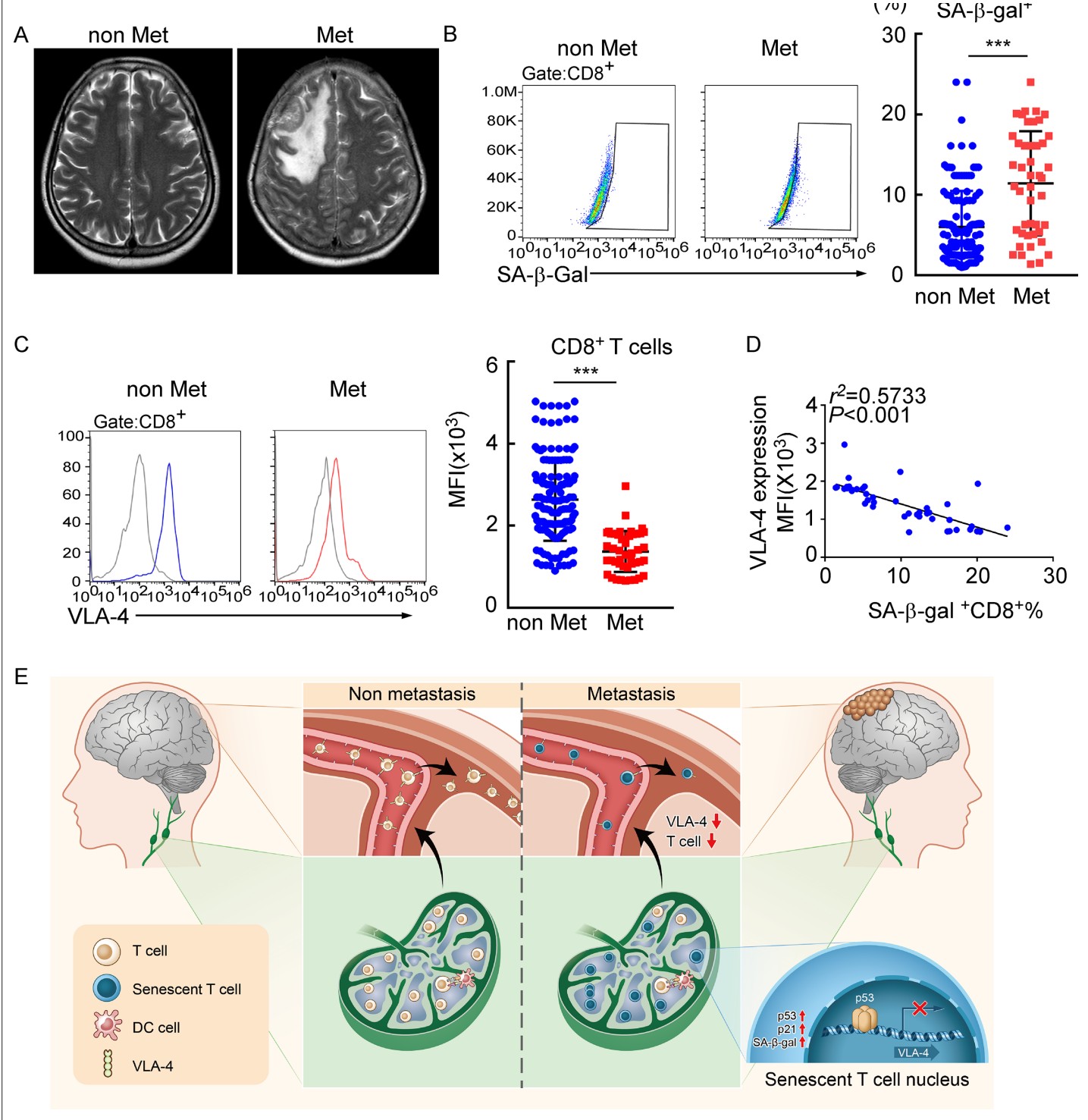

**Figure 7.** Meningeal CD8[+] T cells exhibit VLA-4 downregulation and senescence in human leptomeningeal metastasis (LM). (**A**) Representative MRI images of patients with (n = 45) or without LM (n = 145). (**B**) Representative images for flow cytometry analysis of CD8[+]SA-β-gal[+] T cells from cerebrospinal fluid (CSF) in patients with nonmalignant neurological diseases (n = 145) or LM (n = 45). Bars correspond to the percentages of CD8[+]SA-β-gal[+] T cells. ***p<0.001 by two-tailed Student's t test. (**C**) Flow cytometry analysis of VLA-4 expression in CD8[+] T cells from CSF in patients of nonmalignant neurological diseases (n = 145) or LM (n = 45). Gray, isotype; blue, CD8[+] T cells in patients of nonmalignant neurological diseases; red, CD8[+] T cells in LM patients. Bars correspond to the mean fluorescent intensity (MFI) of VLA-4 in CD8[+] T cells. ***p<0.001 by two-tailed Student's t test. (**D**) The correlation between the percentages of SA-β-gal[+]CD8[+] T cells and VLA-4 levels in CD8[+] T cells of LM patients (n = 45, the Pearson's correlation coefficient r² value and the p-value are shown). (**E**) Schematics highlighting the major findings of this study.

*Figure 7 continued on next page*

*Figure 7 continued*

The online version of this article includes the following figure supplement(s) for figure 7:

**Figure supplement 1.** The relationship between VLA-4 downregulation and senescence in meningeal CD8+ T cells in human leptomeningeal metastasis.

correlated with VLA-4 levels in CD8+ T cells of LM patients (**Figure 7D**). Collectively, these results indicated the downregulation of VLA-4 and senescence in CD8+ T cells in LM (**Figure 7E**).

## Discussion

Extracranial immune cells were traditionally believed to be absent in the normal CNS due to the BBB. The recent identification of functional lymphatic vessels in brain has shifted the paradigm from immunological privilege to a distinct immune response in CNS diseases (**Louveau et al., 2018**; **Ahn et al., 2019**; **Forrester et al., 2018**). LM is associated with one of the worst clinical outcomes of malignancies. However, very little is known about the immune response to LM (**Boire et al., 2020**). Here, we uncovered that tumor-specific CD8+ T cells are generated in dCLNs and play a central role in controlling LM. Moreover, inducing immunosenescence of CD8+ T cells is essential for tumor cells to escape meningeal immune defenses and successfully establish clinical lesions in the leptomeningeal space.

During inflammation, apart from TCR signaling, T cells can also be activated in a T cell receptor-independent and cytokine-dependent manner, which is called 'bystander effect' (**Kim and Shin, 2019**). In the immunocompetent mouse model of breast cancer LM that we constructed, we provided clear evidences that tumor-specific CD8+ T cells can be generated in dCLNs and recruited to leptomeninges by VLA-4. T cells in dCLNs of mice bearing EO771-OVA LM showed specific response to OVA. In addition, OVA-specific T cells underwent antigen-specific activation and proliferation in vitro in response to EO771-OVA injection, indicating that dCLNs generate tumor-specific CD8+ T cells against LM, but not 'bystander activation.' Consistent with our results, previous studies have shown that activation of CNS-specific T cells in cervical lymph nodes can directly mediate the neuroinflammation observed in EAE (**Furtado et al., 2008**). Therefore, our data indicated that the tumor-specific CD8+ T cells primed in dCLNs play an important role in controlling LM.

In brain tumors such as glioblastoma and brain metastases, the infiltrating T cells are dysfunctioned due to tolerance and exhaustion induced by the immunosuppressive tumor microenvironment (**Farber et al., 2016**; **Woroniecka et al., 2018**). Emerging evidence suggests that immunosenescence, which is distinct from exhaustion, is an important state of T cell dysfunction and responsible for immunosuppression in the tumor microenvironment (**Nikolich-Žugich, 2014**). In recent studies, T cells undergo senescence in the normal aging process or in the patients with under chronic infections and cancers (**Zhao et al., 2020**). T cell senescence has also been found to be strongly induced in several types of malignancies, including lung cancer (**Chen et al., 2014**), ovarian cancer (**Wu et al., 2017**), and melanoma (**López-Otín et al., 2013**) through MAPK signaling (**Wang et al., 2002**). MAPK/p38 signaling is essential for activating the cell cycle regulatory molecules p53, p21, and p16, which might inhibit cell cycle progression to slow or completely arrest DNA replication and induce cell senescence (**Montes et al., 2008**). However, whether senescence contributes to immunosuppression in CNS metastasis remained elusive. In our study, we found that T cells in dCLNs from LM mice exhibited senescent features, including elevated expression of p53 and p21, and increased levels of secreted senescence-associated beta-galactosidase. Previous studies have indicated that increased senescent T cells resulted in T cell proliferation arrest and killing capacity defect (**Ye and Peng, 2015**; **Appay et al., 2000**). Here, we advanced this emerging concept by showing that the recruitment of T cells to meninges is impaired as a result of the suppression of VLA-4. VLA-4 is responsible for CNS tropism of T cells and VLA-4 neutralization can inhibit the homing and infiltration of antigen-specific T cells in cerebral autoimmune models (**Burkly et al., 1994**; **Zundler et al., 2017**). VLA-4 blockade in WT mice impedes T cell trafficking to meningeal lymphatics and aggravates LM. Collectively, T cell senescence impairs the trafficking of T cells, leading to the failure of tumor control.

Previous studies have demonstrated that induction of p53 plays a direct role in cellular senescence. Its activation primarily responds to the DNA damage response (DDR) caused by telomere attrition, oxidative, or oncogenic stress (**Mijit et al., 2020**; **Kumari and Jat, 2021**). Also, several p53 targets

and regulators have been associated with induction of senescence (*Feng et al., 2011*). In our study, we identified a specific p53 binding site within the *Itga4* promoter and found that the mutation of specific p53 binding site relieved its repression on the VLA-4 transcription. Consistently, in *Trp53$^{-/-}$* mice, the senescence of CD8$^+$ T cells was inhibited as a result of p53 deficiency, leading to upregulation of VLA-4 and enhanced trafficking ability to meninges. Therefore, p53-deficient T cells can efficiently inhibit tumor growth compared with WT T cells. In addition, in line with our study, previous studies showed that p53-deficient T cells exhibited decreased apoptosis (*Madapura et al., 2012*) and enhanced proliferation in T cells (*Banerjee et al., 2016*), supporting that p53-deficient T cells have higher anti-tumor effector function.

Collectively, our findings revealed that tumor-specific immunity originated from draining dCLNs is essential for restraining LM. Senescence signals inhibit trafficking of CD8$^+$ T cells from dCLNs to meninges and therefore promote the progression of LM.

## Materials and methods

### Mice

6–8-week-old female C57BL/6 mice, BALB/c mice, and nude mice were purchased from the Laboratory Animal Center of Sun Yat-Sen University. *Rag2$^{-/-}$* mice [B6(Cg)-Rag2$^{tm1.1Cgn}$/J, 008449] were purchased from the Jackson Laboratory. *Trp53$^{-/-}$* mice, OT-1 mice on a fully C57BL/6 background were obtained from Shanghai Model Organisms Center Inc (Shanghai, China). All mice were bred and maintained in the specific-pathogen-free (SPF) animal facility of the Laboratory Animal Center of Sun Yat-Sen University. Mice were randomized at the beginning of each experiment, and experiments were not blinded. All procedures were approved by the Animal Care and Use Committee of Sun Yat-Sen University under animal protocol 2018-000095 and 2021-000768.

### Cell culture and treatment

The murine breast cancer cell lines, EO771 cells, were obtained from CH3 Biosystems, NY. Murine 4T1, Lewis Lung Cancer (LLC) cell lines were purchased from ATCC. All the cells and their derivatives were cultured in DMEM or RPIM 1640 with 10% fetal bovine serum (FBS), 2 mM L-glutamine, and 100 units/mL penicillin-streptomycin (all from Gibco). All the cell lines were tested negative for mycoplasma contamination. EO771, 4T1, and LLC cells were forced expressed with firefly luciferase. EO771 cells were transduced with the viral vectors of ovalbumin (EO771-OVA, GenePharma) or glycoprotein B (EO771-gB, Guangzhou Ige Biotechnology) (multiplicity of infection [MOI] of 10) overnight at 37°C with 5 μg/mL polybrene (GenePharma). The established cells were selected by 2 μg/mL puromycin (Sigma).

### Establishment of leptomeningeal metastasis model

An lLM model was generated as previously described with modifications (*Boire et al., 2017*; *Fults et al., 2019*; *Chi et al., 2020*). For establishing breast cancer LM model, leptomeningeal derivative cell line was selected and injected intracardially into C57BL/6 mice. In detail, $2 \times 10^4$ EO771 cells, $2 \times 10^4$ 4T1 cells, and $2 \times 10^4$ LLC cells transduced with lentivirus with forced expression of firefly luciferase (GenePharma) were suspended in 10 μL of PBS and then injected into the cisterna magna of recipient mice. Tumor burdens were monitored by Bioluminescent (BLI). Mice were sacrificed when LM were detected by BLI, or the clinical signs of brain metastasis, including primary CNS dysfunction, weight loss, and behavioral abnormalities, were shown. Then, tumor cells were collected from meninges, selected by puromycin, and reinjected to the second recipient mice. The operations described above were repeated three times to derive intermediate cell line that could survive within the CSF. $1 \times 10^5$ intermediate EO771-luc, 4T1-luc, or LLC-luc cells were inoculated into the intracarotid artery of mice. After LM were detected, mice were sacrificed and tumor cells from meninges were isolated and cultured in vitro, which were identified as LM-phenotype cell line. $1 \times 10^5$ LM-phenotype cells were injected into intracarotid artery of mice to establish the LM model and tumor burden was monitored by BLI and MRI.

### IVIS Lumina imaging

Formation of LM were monitored by IVIS Lumina imaging. Before imaging, mice were anesthetized with ketamine/xylazine injection and injected with d-luciferin (300 mg/kg). After 10 min, mice were

imaged with Xenogen IVIS Lumina system (Caliper Life Sciences). Images were analyzed using Living Image software v.3.0 (Caliper Life Sciences) and BLI flux (photons/s/cm$^2$/steradian) was calculated.

## MRI

MRI experiments were performed on Aspect M3 (1.05 Tesla, Aspect Imaging). Animals were anesthetized with ketamine/xylazine injection throughout the imaging procedure. T1-weighted SE images (TR = 0.6 s; TE = 23 ms) were taken with or without a bolus of 0.5 mmol/kg Gd-DTPA(intravenously, 12.5 min in circulation) (*Brandsma et al., 2004*).

## Immunofluorescence and immunohistochemistry

Paraffin-embedded samples were sectioned at 4 µm thickness. Sections were de-paraffinized, rehydrated, and boiled in a pressure cooker for 2 min in 10 mM citrate buffer (pH 6.0) for antigen retrieval. Then sections were blocked in PBS containing 5% bovine serum albumin (BSA) for 15 min at room temperature. For immunofluorescence assay, sections were incubated with primary antibodies specific for firefly luciferase (Abcam, Cat# ab185924, 1:100) overnight at 4°C and subsequently incubated with Alexa Fluor-488, 555, or 647 conjugated secondary antibodies (Thermo Fisher Scientific, Cat# A32731, A32727, A-21247, 1:300) for 1 hr at room temperature. Cells were counterstained with DAPI. Images were obtained by laser scanning confocal microscopy. For immunohistochemistry assay, sections were incubated with antibodies specific for luciferase (Abcam, Cat# ab185924, 1:100) overnight at 4°C, and later were detected by DAB (Dako) according to the manufacturer's instructions.

## Transcriptomic analysis

Cells cultured in T25 flask at 75% confluency were collected in TRIzol Reagent (Invitrogen) for RNA extraction. mRNA purified from cancer cells was used for library construction with TruSeq RNA Sample Prep Kit v2 (Illumina) following the manufacturer's instructions. Samples were barcoded and run on a Hiseq 2000 platform in a 50 bp/50 bp paired-end run using the TruSeq SBS Kit v3 (Illumina). An average of 40 million paired reads were generated per sample. FASTQ files from RNA-Seq results were quality assessed by FastQC v0.11.3. Raw reads were mapped to human genome hg19 (GRCh37, February 2009) or mouse genome mm10 (GRCm38, December 2011) using STAR2.3.0e (*Engström et al., 2013*) with standard settings for paired-end sequencing reads. On average, 84% of raw reads were uniquely mapped. Mapped reads were counted to each gene by HTSeq v0.5.4 with default settings. Differential gene expression analysis was performed following the instructions of "DESeq2" package deposited in Bioconductor.

## *In vivo* administration of antibodies

Depleting antibodies to CD4 [GK1.5] and CD8 [53.6.72] were administered intraperitoneally (i.p.) (0.25 mg/mouse) on days 1, 4, and 6. Blocking antibody to VLA-4 [PS/2] was administered by i.p. injection (0.25 mg/mouse) on days 0, 1, and 2 since the day of tumor cell injection. Rat-anti-mouse IgG (Cat# BE0090) was used as control antibody. All antibodies were obtained from BioXCell.

## Evans blue injection and detection

Mice were anesthetized by ketamine/xylazine i.p. injection, and then 5 µL of 10% Evans blue (Sigma-Aldrich) was delivered into the cisterna magna via intracerebroventricular (i.c.v) injection. Then, 30 min after injection, the CNS draining lymph nodes were dissected for assessment of Evans blue.

## Isolation of immune cells from secondary lymphoid organs

Briefly, mice were anesthetized with ketamine/xylazine injection. Spleen, inguinal lymph nodes, or dCLNs were isolated, smashed, and filtered through a 40 µm filter to obtain single-cell suspension. For acquisition of meninge immune cells, transcardial perfusion with 30 mL of PBS via intracardiac puncture was performed before meninge isolation (*Louveau et al., 2018*).

## Adoptive T cell transfer

For adoptive T cell transfer therapy, dCLNs from mice injected with EO771-luc cells were smashed and filtered through a 40 µm filter to obtain single-cell suspension. CD4$^+$ and CD8$^+$ T cells were purified by magnetic-activated cell sorting (Miltenyi, Cat# 130-117-043, 130-096-495). Cell populations were

confirmed to be >90% purity by flow cytometric analysis. Thereafter, $5 \times 10^5$ CD4+ and CD8+ T cells were injected into $Rag2^{-/-}$ mice via caudal veins, respectively.

## Flow cytometry

Cells collected from secondary lymph organs or CSF of patients were stained with fluorescent-conjugated antibody: fixable viability dye (Thermo Fisher Scientific, Cat# 65-0866-14), CD45 (BioLegend, Cat# 103132), CD3 (BioLegend, Cat# 100213), CD8 (Thermo Fisher Scientific, Cat# 53-0081-82, 17-0088-42), CD4 (Thermo Fisher Scientific, Cat# 62-00420-80), CD11b (BioLegend, Cat# 101222), Ly6G (BioLegend, Cat# 127613), Ly6C (BioLegend, Cat# 128017), CD11c (BioLegend, Cat# 117322), H-2Kb bound to SIINFEKL (BioLegend, Cat# 141605), OVA pentamer (ProImmune, Cat# F93-2A-G), SA-β-gal (Dojindo, Cat# SG03), and CD49d (Thermo Fisher Scientific, Cat# 12-0492-81, 16-0492-85) for 30 min at 4°C; primary antibodies: luciferase (Abcam, Cat# ab185924, 1:100), cleaved caspase-3 (Cell Signaling Technology, Cat# 9661, 1:200), and Ki-67 (Thermo Fisher Scientific, Cat# 14-5698-80, 1:50). For intracellular staining, cells were pretreated with Foxp3/Transcription Factor Fixation/Permeabilization kit (eBioscience, Cat# 00-5521-00) according to the manufacturer's instructions. Flow cytometry was performed on Attune NxT Flow Cytometer (Thermo Fisher Scientific) and analyzed using FlowJo software.

## *Ex vivo* activation of T cells

PBS, EO771, and EO771-gB and EO771-OVA were injected into C57BL/6 mice via cisterna magna. 7 days after tumor cell injection, DCs were isolated from murine lymph nodes using magnetic beads according to the manufacturer's instructions (Miltenyi Biotec, Cat# 130-100-875). OT-1 T cells were isolated from peripheral blood of OT-1 mice and labeled with 0.5 μM CFSE (Thermo Fisher, Cat# C34554) for 15 min at 37°C. After purification, DCs ($5 \times 10^4$) and CFSE-labeled OT-1 T cells ($10^5$) were incubated in flat 96-well plate for 60 hr. Subsequently, OT-1 T cells were harvested for flow cytometry and IFN-γ ELISpot assay.

## IFN-γ ELISpot assay

ELISpot assays were performed using a mouse IFN-γ ELISpot kit according to the manufacturer's procedure (DAKAWE, China). The harvested OT-1 T cells were added to ELISpot plates and cultured overnight at 37°C. Plates were then washed four times with washing buffer and incubated with biotinylated detection antibody for IFN-γ (DAKAWE) for 1 hr at 37°C. Following four additional washes with washing buffer, the plates were incubated with avidin–horseradish peroxidase (HRP) (DAKAWE) for 1 hr at 37°C. Plates were then washed four more times with washing buffer and two washes with PBS. After 15 min incubation with AEC substrate (BD), the reaction was stopped. The plates were washed with deionized water and dried overnight before membrane removal. Spots were counted using an ELISpot reader (ImmunoSpot S6 ULTRA-V, Cellular Technology, Cleveland, OH).

## Lymphatic vessel ligation

Mice were anesthetized by ketamine and xylazine, and the skin of the neck was shaved and cleaned with iodine and 70% ethanol. A midline incision was made on the neck 5 mm above the clavicle. The sternocleidomastoid muscles were retracted, and the dCLNs were exposed on both sides. Ligation of the afferent lymphatic vessels was performed with 10-0 synthetic, nonabsorbable sutures. Sham surgeries consisting of the skin incision and retraction of the sternocleidomastoid muscle were performed on the control mice. The incision was sutured and the mice were allowed to recover on a heat pad until fully awake.

## Two-photon microscopy

Before two-photon microscopy, $1 \times 10^6$ CD8+ T cells from dCLNs of mice injected with parental or LM cells were isolated, labeled by 0.5 μM CFSE (Thermo Fisher, Cat# C34554) for 15 min at 37°C, and transferred to recipient mice via caudal veins. 24 hr after T cell transfusion, mice were anesthetized, and thinned skull windows were prepared. Mice were injected with 100 μL rhodamine dextran solution (100 mg/mL, Sigma, R9379) where indicated to visualize blood vessels. Images were captured in z-stacks of 10–30 planes (1 μm step size) using an Olympus FVMPE-RS two-photon microscope.

## Whole-mount meninge preparation

Anesthetized mice were euthanized, brains were sterilely dissected, and placed in ice-cold sterile PBS. Meninges within the skullcap were fixed in 4% paraformaldehyde (PFA) overnight and separated from the skullcap. Then the meninges were incubated in the block-perm buffer containing 2% normal goat serum, 1% BSA, 0.1% Triton X, and 0.05% Tween for 1 hr at room temperature with gentle rocking. The primary antibody anti-mouse CD8 (Novus, Cat# ABX-160A, 1:100), anti-lyve-1 (R&D Systems, Cat# AF2125), and the secondary antibody Alexa Fluor 488, 555 rabbit anti-mouse IgG, Alexa Fluor 555 rabbit anti-goat IgG (Life Technologies) were employed for CD8$^+$ T cell and lymphatic vessel staining. The meninges were mounted with Prolong Gold with DAPI (Molecular Probes).

## DTC detection with luciferase assay

100 µL blood was centrifuged at 92 × $g$ for 5 min, and plasma was discarded and the sediment were subsequently for flow cytometry and luciferase assay. For flow cytometry, cells were stained with anti-firefly luciferase (Abcam, Cat# ab185924, 1:100) and anti-CD45 antibodies (BioLegend, Cat# 103132). For luciferase assay, cell lysis buffer (200 µL) (Beyotime) was added to resuspend cells, incubated at room temperature for 8–10 min with occasional shaking, then centrifuged at 12,000 × $g$ for 5 min. The supernatant was aspirated into another tube and 100 µL of luciferin working solution (Beyotime) was added to the sample, reacting with luciferase of DTCs of EO771-luc. Immediately, the RLU in each sample was assayed by luminometer (Infinite M200 Pro, Tecan). To set up standard curve, the freshly harvested EO771-luc cells were counted, and 0, 5, 10, 20, 30, 40, 50, and 60 cells were added to tubes, respectively. RLU was assayed and the derived equation was used as standard curve to calculate the DTC numbers from mice with metastases.

## Cell adhesion assays

According to Sasaki et al., 96-well-ELISA plates were coated with 10 µg/mL of mouse VCAM-1 recombinant protein or heat-denatured BSA. T cells harvested and suspended in binding buffer (0.5% BSA, 2 mM CaCl$_2$, 2 mM MgCl$_2$ in PBS) were subsequently added to the plate. For blocking experiments, cells suspended in binding buffer were pretreated with 20 µg/mL of α-CD49d mAbs (PS/2) for 15 min at 37°C, and then added to the plate. Plates were centrifuged at 500 rpm for 1 min and were placed at room temperature with gentle shaking for 30 min for cell adhesion. The plate was then gently washed three times using binding buffer, and the number of adherent cells was counted by flow cytometry. Percent adhesion was calculated as the (number of adherent cells to VCAM-1 - number of adherent cells to heat-denatured BSA)/Number of total input cells.

## Cell migration assays

Murine brain microvascular endothelial cells (MBMECs) were isolated as previously described (*Meena et al., 2021*; *Ruck et al., 2014*). In brief, mice were sacrificed and meninge-free forebrains were collected, minced, then digested with 10 mg/mL of collagenase II and 1 mg/mL DNase (Worthington Biochemical) in DMEM in a shaker for 1 hr at 37°C. The digested tissues were suspended with 20% BSA solution prepared in DMEM to remove myelin. The pellets were further digested by 1 mg/mL of collagenase/dispase and 1 mg/mL DNase (Roche Applied Science) in DMEM for 1 hr at 37°C. The microvessels obtained from the pellets after digestion were separated on a 33% continuous Percoll gradient (700 g, 10 min). The microvessel compartments were washed twice in DMEM and cultured in endothelial cell medium. Five days after isolation, MBMECs were trypsinized and seeded onto precoated Transwell inserts (pore size 3 µm; Corning) at 2 × 10$^4$ cells per insert. When MBMECs reached confluence, 2 × 10$^5$ T cells per insert were added directly on top of MBMECs. After 24 hr, migrated cells were collected from the lower chamber, while nonmigrating cells remained in the upper chamber.

## Western blot

dCLN CD8$^+$ T cells were washed with PBS and lysed with RIPA buffer containing proteinase inhibitor. Lysates were centrifuged for 25 min at 4°C at 12,000 × $g$. Protein extracted from the cells were fractionated by 10% SDS–polyacrylamide gels and further transferred to polyvinylidene difluoride membranes. Membranes were blocked with TBS/0.05% Tween-20/5% skim milk and then incubated with primary antibodies against p53 (ProteinTech, Cat# 10442-1-AP, 1:1000), p21 (ProteinTech,

Cat# 27296-1-AP, 1:1000), and GAPDH (ProteinTech, Cat# HRP-60004, 1:10,000) at 4°C overnight. Membranes were washed three times with TBST and then incubated with peroxidase-conjugated secondary antibody (Cell Signaling Technology, Cat# 7074S, 1:3000). The antigen-antibody reaction was visualized by enhanced chemiluminescence assay (ECL, Thermo Fisher Scientific).

## Cellular senescence assay

For senescence-associated β-galactosidase (SA-β-gal) staining, dCLN CD8+ T cells were isolated, fixed, and stained for SA-β-gal at 37°C overnight by Senescence-associated β-Galactosidase Staining Kit (Beyotime, Cat# C0602) according to the manufacturer's instruction. For quantification of SA-β-gal-positive cells, images were randomly taken at ×40 magnification (BX-63, Olympus) and then analyzed manually with ImageJ.

## Luciferase reporter assay

The luciferase assay was performed using luciferase assay reagent according to the manufacturer's instructions. Briefly, a fragment spanning from –2000 to +100 relative to the TSS of the murine VLA-4 genomic sequence was fused to pGL3-Basic vector to generate VLA-4 WT (–2000/+100)-luc. The C/G to A/T mutations at the consensus p53-binding site were introduced by site-directed mutagenesis (from 5'-GGAGCCC-3' to 5'-GGATAAA-3'). For luciferase reporter experiments, a murine T lymphoma cell line EL4 was transfected with VLA-4-wild type-luc or VLA-4-mutant-luc reporter and pRL-TK-Renilla for 24 hr using a Bio-Rad Gene Pilser Xcell Electroporation System. Afterward, the cells were transduced with lentivirus with p53 overexpression or empty vector. After 24 hr transfection, cells were lysed and analyzed with the Dual-Luciferase reporter assay system (Promega, Cat# E1960). Renilla Luciferase (R-luc) was used to normalize firefly luciferase (F-luc) activity to evaluate reporter translation efficiency.

## Chromatin immunoprecipitation assay (ChIP)

The ChIP assay was performed using a ChIP Assay Kit (Millipore) according to the manufacturer's instructions. Briefly, dCLN CD8+ T cells were isolated and fixed in 1% formaldehyde for 10 min at room temperature. Fixed cells were washed and then lysed in ChIP lysis buffer. The whole-cell extracts were then sonicated for 10 cycles of 10 s on /20 s off and 50% AMPL with Sonics VCX130 (Sonics & Materials, Inc, Newtown). Antibodies directed against p53 (ProteinTech, 10442-1-AP, 1 µg) or rabbit IgG (ProteinTech, Cat# B900610, 1 µg) were used. The precipitated DNA was subjected to PCR amplification. The primer sequences used in ChIP assays:

> *Itga4*: forward (5'>3') TCTTCTCAGAGTGTGTGGA;
> reverse (5'>3') GAGCACCCAGCAACATTT.
> *Gapdh*: forward (5'>3') GCCCTGCTTATCCAGTCCTA;
> reverse (5'>3') GGTCCAAAGAGAGGGAGGAG.
> *Cdkn1a*: forward (5'>3') TAGCTTTCTGGCCTTCAGGA;
> reverse (5'>3') TGGGTATCATCAGGTCTCCA.

## Patients and tissue samples

CSF samples were obtained by lumbar puncture or extraventricular drain from 145 cases of nonmalignant neurological diseases (45 cases from brain injury, 25 cases from neurodegenerative diseases, 50 cases from cerebrovascular diseases, and 25 cases from benign primary tumors) and 45 cases of brain metastasis, including 6 cases of breast cancer, 35 cases of lung cancer, and 4 cases of gastrointestinal cancer at Sun Yat-Sen Memorial Hospital, Sun Yat-Sen University (Guangzhou, China) between 2017 and 2020. CSF sample were centrifuged at 450 × *g* for 8 min, and the cells of the CSF sediment were collected and preserved in cryovials with the 1 mL of freezing medium (90% FBS supplemented with 10% DMSO) in liquid nitrogen for future flow cytometry analysis. To minimize bias, samples were blinded to laboratory personnel. All samples were collected with informed consents, and all related procedures were performed with the approval of the internal review and ethics board of Sun Yat-Sen Memorial Hospital under protocol 2020-136.

## Statistics

The number of events and information about the statistical details and methods are indicated in the relevant figure legends. Data are expressed as the mean ± SD and were analyzed using GraphPad

Prism 8.0. Two-tailed Student's *t*-tests were used to identify significant differences between two groups. One-way ANOVA with Tukey's multiple-comparison test were used for comparison of more than two groups. Pearson's correlation was used to assess the relationship between VLA-4 expression and the activity of SA-β-gal in CD8$^+$ T cells in the CSF of patients with brain metastases. Kaplan–Meier survival curves were plotted and log-rank test was done. $p < 0.05$ was considered statistically significant. The experiments were not randomized, and the investigators were not blinded to allocation during experiments and outcome assessment, except for when noted otherwise.

## Acknowledgements

This work was supported by grants from the National Key Research and Development Program of China (2021YFA1300502 to SS), the Natural Science Foundation of China (91942309 to SS, 92057210 to SS, 82222029 to DH, 82071754 to YW, 82271793 to YW, 81971481 to QZ, 82173064 to QZ, 81602205 to BL, 82002786 to LY, 82003859 to MN), the Natural Science Foundation of Guangdong Province (2020A1515011458 to DH, 2022B1515020023 to QZ, 2021A1515010230 to LY), Science and Technology Program of Guangzhou (202103000070 to SS, 202201020467 to QZ), Sun Yat-Sen Projects for Clinical Trials (SYS-C-202003 to YW), Fundamental Research Funds for the Central Universities (22ykqb01 to QZ).

---

## Additional information

### Funding

| Funder | Grant reference number | Author |
|---|---|---|
| National Key Research and Development Program of China | 2021YFA1300502 | Shicheng Su |
| Natural Science Foundation of China | 91942309 | Shicheng Su |
| Natural Science Foundation of China | 92057210 | Shicheng Su |
| Natural Science Foundation of China | 82222029 | Di Huang |
| Natural Science Foundation of China | 82071754 | Ying Wang |
| Natural Science Foundation of China | 82271793 | Ying Wang |
| Natural Science Foundation of China | 81971481 | Qiyi Zhao |
| Natural Science Foundation of China | 82173064 | Qiyi Zhao |
| Natural Science Foundation of China | 81602205 | Bingxi Lei |
| Natural Science Foundation of China | 82002786 | Linbing Yang |
| Natural Science Foundation of China | 82003859 | Man Nie |
| Natural Science Foundation of Guangdong Province | 2020A1515011458 | Di Huang |
| Natural Science Foundation of Guangdong Province | 2022B1515020023 | Qiyi Zhao |

| Funder | Grant reference number | Author |
| --- | --- | --- |
| Natural Science Foundation of Guangdong Province | 2021A1515010230 | Linbing Yang |
| Science and Technology Program of Guangzhou | 202103000070 | Shicheng Su |
| Science and Technology Program of Guangzhou | 202201020467 | Qiyi Zhao |
| Sun Yat-Sen Projects for Clinical Trials | SYS-C-202003 | Ying Wang |
| Fundamental Research Funds for the Central Universities | 22ykqb01 | Qiyi Zhao |

The funders had no role in study design, data collection and interpretation, or the decision to submit the work for publication.

### Author contributions

Jiaqian Li, Investigation, Methodology, Writing – original draft; Di Huang, Funding acquisition, Validation, Investigation, Methodology, Writing – original draft, Writing – review and editing; Bingxi Lei, Validation, Investigation; Jingying Huang, Investigation, Writing – original draft; Linbing Yang, Funding acquisition, Writing – review and editing, Verification of research outputs; Man Nie, Funding acquisition, Writing – review and editing, Verification of research outputs; Shicheng Su, Conceptualization, Supervision, Writing – original draft, Project administration, Writing – review and editing; Qiyi Zhao, Supervision, Investigation, Methodology; Ying Wang, Conceptualization, Formal analysis, Funding acquisition, Investigation, Methodology, Project administration

### Author ORCIDs
Jiaqian Li http://orcid.org/0000-0001-7162-8960
Ying Wang http://orcid.org/0000-0002-0413-831X

### Ethics

Human subjects: All samples were collected with informed consents, and all related procedures were performed with the approval of the internal review and ethics board of Sun Yat-Sen Memorial Hospital under protocol 2020-136.
All mice were bred and maintained in the specific-pathogen-free (SPF) animal facility of the Laboratory Animal Center of Sun Yat-Sen University. All procedures were approved by the Animal Care and Use Committee of Sun Yat-Sen University under animal protocol (2018-000095 and 2021-000768).

### Decision letter and Author response
Decision letter https://doi.org/10.7554/eLife.83272.sa1
Author response https://doi.org/10.7554/eLife.83272.sa2

## Additional files

### Supplementary files
• MDAR checklist

### Data availability
All data generated or analyzed during this study are included in the manuscript and supporting file; Source Data files have been provided for Figure 1H, Figure 4G and Figure 4—figure supplement 1I.

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
