## [Editor Report]

This study address an exciting area which remains less explored. Overall, the findings presented here provides immediate impact to cancer metastasis and how the interplay between tumor cells and immune cells can affect this process.

---

## [Decision Letter]

**Decision letter after peer review:**

[Editors’ note: the authors submitted for reconsideration following the decision after peer review. What follows is the decision letter after the first round of review.]

Thank you for submitting the paper "VLA-4 suppression by senescence signals regulates meningeal immunity and leptomeningeal metastasis" for consideration by *eLife*. Your article has been reviewed by 2 peer reviewers, including Ping-Chih Ho as the Reviewing Editor and Reviewer #1, and the evaluation has been overseen by a Senior Editor.

Comments to the Authors:

We are sorry to say that, after consultation with the reviewers, we have decided that this work will not be considered further for publication by *eLife*.

This decision is based on the evaluation from reviewers since critical issues have been raised.

*Reviewer #1:*

Overall, it is an interesting paper to address how T cells fail to prevent leptomeningeal metastasis. The topic is timely and interesting. However, several key weakness of this manuscript can be identified as below:

1. T cell activation part present in figure 3 is problematic. The transferred of OT-1 cells in the design does not support dCLN can facilitate generation of tumor antigen-specific T cells. It suggests that OT-1 T cells in dCLN lost cytokine production in OVA-expressing tumor-bearing mice. I would suggest to rewrite this and re-perform the experiment for strengthening the conclusion.

2. The contribution of VLA4 is not fully examined in this study despite that a strong correlation is presented.

3. How p53 is activated in CD8 is not unknown and whether p53 targeting can improve the metastasis control is not clear.

4. Moreover, the involvement of CD8 on controlling metastasis in metastatic locations or affecting dissemination of cancer cells is not clear. This is the key message of this paper. Thus, it should be addressed properly.

Although the message is interesting, I do suggest the authors to address immunology parts, especially T cell activation and the contribution of CD8 T cells on hampering leptomeningeal metastasis, in more details. Moreover, a direct examination the role of VLA4 expression in T cells on hampering metastasis is needed.

*Reviewer #2:*

In the submitted manuscript, Li et al. propose an interesting hypothesis that meningeal CD8^+^ T cells enter senescence in a p53-dependent manner in leptomeningeal metastasis (LM). While the observations are intriguing, the manuscript is not prepared carefully and lacks information critical for rigor and reproducibility. The use of a single mouse model severely limits the generalizability of the claims. Moreover, the mouse experiments are underpowered in all cases. The immunological observations are not internally consistent. Finally, the clinical validation materials are derived from what they call 'brain metastasis' (parenchymal metastasis), a completely different entity from leptomeningeal metastases, and these findings should not be generalized to LM.

Major Points (A-E):

A. Many of the claims and conclusions must be reconsidered as the primary observations are derived from a single cell line. This is not sufficient to make generalized conclusions that are stated in the Results section, given the lack of similar observations in human breast cancer LM patients. The community standard for such claims is at least two cell lines (and corresponding models); three would provide even greater certainty. The model itself is also inadequately characterized:

– Quantification of disease burden is not consistent throughout the manuscript: The authors state that they quantified LM tumor volume from MRI experiments using the formula commonly used to quantify subcutaneous tumors: length x width2 x 0.52. Given that LM forms sheet-like structure on the surface of the brain, rather than a sphere, this formula is not appropriate. Later in the manuscript, they use (2D) area rather than volume. This discrepancy must be addressed.

– New models of metastasis generated through iterative in vivo selection are typically introduced with the corresponding transcriptome. This is a great service to the larger metastasis community. It would also go a long way toward overcoming the use of a single cell line in the manuscript. Is it possible for the authors to characterize the bulk transcriptome of Parental, Intermediate and LeptoM cells?

– Tumor cells in Figure 2B are not really apparent. Also, different brain areas seem to be showed. Images and insets from the same anatomic locations would be helpful.

B. Certain of the Immunologial observations presented are either internally inconsistent or are inconsistent with clinical data.

– Authors use gP as a control for OVA experiments. Cancer cells overexpressing gB (human?) were introduced into wild type mice with adoptively transferred OT-I CD8^+^ cells, then dcLN CD8^+^ cells were isolated and challenged in vitro with corresponding peptides. No response is observed when CD8^+^ cells from E0771 gB mice are cultured with the peptides. In figure 2F the authors show that dcLN CD8^+^ cells from E0771 primed mice are able to control the growth of cancer and provide survival benefit, i.e. the cells in dcLNs recognize E0771 cancer cells to some extent. How would authors explain and address this discrepancy – the lack of gB specific CD8 T cells? Can authors isolate cells from E0771 WT mice and expose them to E0771 WT lysate to estimate the proportion of cancer-specific cells?

– In Figure 4, the authors are showing drop in meningeal T cell numbers in the presence of cancer. LM is known to induce pleocytosis, this would not be reflected in relative numbers and thus absolute CD3 and CD8 numbers should be provided. What immune cell types are taking over meninges in the presence of cancer?

– Authors claim that in CD8^+^ T cells in P53 deficient mice do not downregulate VLA4 and likely do not undergo senescence. Can authors reproduce the results from Figure 5 in P53 deficient mice? Does cancer grow less in these mice? BLI signal should be quantified.

C. Setting aside the use of a single mouse model, there are serious issues with the work's rigor and reproducibility:

– Some of the antibody dilutions (1:00) seem incorrect.

– For Figure 5 source data, full membrane scans should be provided. The images that are shown are still somewhat cropped.

– Why were the LeptoM cells established in Rag2-/- mice (line 306)?

– Authors claim they performed whole mount stainings of meninges, but such images are nowhere to be found.

– In methods, authors say that the data are representative of at least three independent experiments with 4-6 mice per experiment. Most experiments in the submitted manuscript show only 5, sometimes 4 animals. Why did the authors not include the other two experiments, i.e. majority of animals? Please, provide data from all performed experiments as dot plots (not bar plots).

– The actual BLI quantification should be shown in main figures.

– In Figure 2, authors state that tumor growth was monitored by BLI, but show pictures of brains ex vivo. It should be clearly stated in the figure legends what signal and at what time point was quantified.

– Additional positive and negative controls for the P53 ChIP in Figure 6 should be provided. It would be also helpful to align the mutant Itga4 promoter sequence below the WT one in Figure 6B.

D. The methods section is incomplete and does not provide sufficient detail for interested readers to reproduce the experiments, and/or for adequate assessment of the experiments presented. Methods need to be carefully checked and all the lacking information regarding reagents and their use should be provided. In particular:

– Details related to senescence analysis with ImageJ are not provided.

– It is unclear how were the in vivo treatments with antibodies performed.

– Gating strategy for tissues used in this manuscript should be shown.

– Authors state that they purified CD4^+^ T cells and confirmed their purity with flow cytometry, but no anti-CD4 antibody is mentioned in the methods.

– OT-I mice are not mentioned in methods.

– Were the transgenic mice maintained as homozygous colonies? This statement should be added into the corresponding methods paragraph.

– Numerous viral vectors (luciferase, ovalbumin, glycoprotein B) are employed; the source and backbone are unclear.

– There is no source, concentration, length of incubation mentioned for several reagents (OVA and gB proteins, CFSE etc.)..

– How were the CSF pellets preserved and stored?

E. The clinical samples do not provide validation for the overall hypothesis:

– Although the authors analyzed an impressive number of clinical samples (Figure 7), the metastatic samples seem to be derived from "brain metastases" and not LM. Given this limitation, the proposed observations from a single mouse model cannot be reflected to human LM.

– An inverse correlation is demonstrated in Figure D; a negative Pearson's correlation coefficient is expected.

– Gating strategy for human samples has to be shown.

---

## [Author Response]

[Editors’ note: the authors resubmitted a revised version of the paper for consideration. What follows is the authors’ response to the first round of review.]

Reviewer #1:Overall, it is an interesting paper to address how T cells fail to prevent leptomeningeal metastasis. The topic is timely and interesting.

We thank the reviewer for the positive comments for our manuscript.

However, several key weakness of this manuscript can be identified as below:1. T cell activation part present in figure 3 is problematic. The transferred of OT-1 cells in the design does not support dCLN can facilitate generation of tumor antigen-specific T cells. It suggests that OT-1 T cells in dCLN lost cytokine production in OVA-expressing tumor-bearing mice. I would suggest to rewrite this and re-perform the experiment for strengthening the conclusion.

We thank the reviewer for the constructive suggestion. We have performed additional experiments to support our conclusion that dCLN can facilitate the generation of tumor antigen-specific T cells. The details are as below:

1) It’s well documented that the generation of tumor antigen-specific CD8^+^ T cells depends on effective tumor antigen presentation by dendritic cells (DCs) in lymph nodes (LNs), which is the fundamental step that launches T cell response against tumor (1,2). Thus, we firstly examined the peptide/MHC class I complex on the cell surface of DCs of mice bearing leptomeningeal metastasis, which indicated the tumor antigen presentation by DCs in LNs. Since deep cervical lymph nodes (dCLNs) could communicate with meningeal lymphatics directly, dCLNs were considered as the tumor draining LNs and inguinal LNs were taken as nondraining LNs. We inoculated EO771 breast tumor cells with ectopic expression of chicken ovalbumin (OVA) into the cisterna magna of C57BL/6 mice and isolated the CD11c^+^ DCs from dCLNs, inguinal LNs and spleen 7 days later. We found that OVA peptide/MHC class I complex, SIINFEKL/H2-Kb, was only detected on CD11c^+^ DCs in dCLNs, but not the ones in inguinal LNs or the spleen (revised Figure 3A, Figure 3—figure supplement 3A, B).

2) Further, to investigate whether dCLNs can give rise to tumor-specific CD8^+^ T cells during tumor dissemination to leptomeningeal space, we injected PBS, EO771 cells (EO771 WT), EO771-OVA and EO771 with an irrelevant antigen glycoprotein B (EO771-gB) into C57BL/6 mice via the cisterna magna. We used H-2Kb-OVA/SIINFEKL pentamer staining to detect OVA-specific CD8^+^ T cells in dCLN after tumor inoculation and observed that a significant proportion of OVA/SIINFEKL pentamer^+^ CD8^+^ T cells was detected in the dCLNs of mice injected with EO771-OVA cells, but not in the mice injected with PBS, EO771 WT or EO771-gB cells (revised Figure 3B, Figure 3—figure supplement 3C). Furthermore, CD11c^+^ DCs were subsequently isolated from draining dCLNs, and then co-cultured with naive OT-1 CD8^+^ T cells (revised Figure 3C). Analysis of OT-1 T cell priming revealed that only DCs from mice receiving EO771-OVA inoculation could expand OT-1 T cells (revised Figure 3D, Figure 3—figure supplement 3D) and induce a significant increase in IFN-γ production (revised Figure 3E, Figure 3—figure supplement 3E) of OT-1 T cells.

3) In addition, surgical ligation of the lymphatic afferent to the dCLNs which could impair the antigen presentation in LNs was performed before EO771-OVA tumor cell inoculation (revised Figure 3F). Determined by the expression of SIINFEKL/H2-Kb complex on the surface of CD11c^+^ DCs, antigen processing (revised Figure 3G, Figure 3—figure supplement 3F) was impaired in the mice receiving surgical ligation. In addition, the proliferation (revised Figure 3H, Figure 3—figure supplement 3G) and IFN-γ production (revised Figure 3I, Figure 3—figure supplement 3H) of OT-1 T cells primed by DCs were diminished.

2. The contribution of VLA4 is not fully examined in this study despite that a strong correlation is presented.

In the previous manuscript, we found that VLA-4 level was downregulated in CD8^+^ T cells in mice injected with LM-phenotype cells and the trafficking capacity of these CD8^+^ T cells was impaired. We also proved that VLA-4 blockade diminished CD8^+^ T cell recruitment to meninges and then aggregated intracranial tumor metastasis in mice injected with parental tumor cells. To further address this question, we have examined the contribution of VLA-4 in T cell trafficking, including adhesion and migration.

1) As previously reported, VLA-4 can bind to vascular cell adhesion molecule-1 (VCAM-1) on the endothelial cells, which was required for T cells to access into central nervous system (CNS) (3,4). To further investigate the role of VLA-4 in T cell adhesion, we isolated CD8^+^ T cells from the dCLNs of mice injected with EO771 LM-phenotype cells (LM-CD8^+^ T cells) and parental cells (Parental-CD8^+^ T cells) and tested their ability to adhere to plate-bound VCAM-1-Ig fusion protein (revised Figure 5G) (5). In line with the differential expression of VLA-4 expression on CD8^+^ T cells, Parental-CD8^+^ T cells exhibited specific adhesion to plate-bound VCAM-1-Ig (74.77±4.27%), whereas, LM-CD8^+^ T displayed only background levels of adhesion (9.96±0.89%) (revised Figure 5G). Moreover, pretreatment of Parental-CD8^+^ T cells with the anti-VLA-4 Ab virtually ablated their ability to adhere to VCAM-1(19.22±3.44%), while LM-CD8^+^ T cells pretreated with the anti-VLA-4 Ab showed no difference in their ability to adhere to VCAM-1 (8.43±0.85%), supporting that downregulation of VLA-4 in LM-CD8^+^ T cells impaired their ability of adhesion (revised Figure 5G).

2) To further investigate the role of VLA-4 in T cell migration, we employed an in vitro blood-brain barrier model to assess the transmigration of T cells. Briefly, primary glial cell cultures were obtained from newborn mouse cerebral cortex and the capillaries were isolated from arterioles and venules of the brain vascular components. Endothelial cells (ECs) were harvested from digested capillaries, and later were plated on the upper side of transwell inserts until they reached confluence about 4–5 days after plating(6,7). LM- and Parental CD8^+^ T cells were added to the top chamber and treated with IgG or VLA4 antibody. We observed that the migration of Parental-CD8^+^ T cells was much stronger than LM-CD8^+^ T cells, which was disrupted by anti-VLA4 treatment (revised Figure 5H).

Collectively, these data suggest that VLA-4 plays an essential role in the adhesion and migration of T cells.

3. How p53 is activated in CD8 is not unknown and whether p53 targeting can improve the metastasis control is not clear.

We thank the reviewer for bringing up this important issue. Previous studies have demonstrated that induction of p53 is pivotal for the establishment of senescence, mainly following its activation by the DNA damage response (DDR) caused by telomere attrition, oxidative or oncogenic stress (8,9). Also, several p53-targets and regulators have been linked to induction of senescence (10). In recent studies, T cells undergo senescence in the normal aging process or in the patients with under chronic infections and cancers (11). Moreover, tumor-derived soluble factors induced T cell senescence through MAPK signaling (12). MAPK/p38 signaling is essential for activating the cell cycle regulatory molecules p53, p21, and p16, which might inhibit cell cycle progression to slow or completely arrest DNA replication and induce cell senescence (13).

To further explore whether p53 targeting can improve the metastasis control, we injected EO771 LM-phenotype or parental tumor cells in wildtype (WT) and *Trp53*^-/-^ mice. In *Trp53*^-/-^ mice, the senescence of CD8^+^ T cells was prevented as a result of p53 deficiency (Figure 6—figure supplement 6A-B), contributing to upregulation of VLA-4 (Figure 6A) and enhanced trafficking ability to meninges (Figure 6—figure supplement 6 C-D). Therefore, leptomeningeal metastatic tumor growth was inhibited in P53 deficient mice determined by BLI signal (Figure 6—figure supplement 6E-F). In addition, previous studies showed that p53 deficient T cells exhibited decreased apoptosis (14) and enhanced proliferation in T cells(15), supporting that p53 deficient T cells have higher anti-tumor effector function. Therefore, targeting p53 in CD8^+^ T cells can improve the metastasis control. As suggested, we have provided further discussion in the revised manuscript to clarify this issue (Line 406-418, Page 15).

4. Moreover, the involvement of CD8 on controlling metastasis in metastatic locations or affecting dissemination of cancer cells is not clear. This is the key message of this paper. Thus, it should be addressed properly.

We thank the reviewer for bringing up this important issue. Indeed, breast cancer is hematogenous spread to the leptomeninges through arterial vessels and venous circulation (16,17). Therefore, we and others (18) established the leptomeningeal derivatives of breast cancer cells by hematogenous dissemination through intracarotid artery injection. In the previous manuscript, we found that the mice with CD8^+^ T cell depletion dramatically increased intracranial tumor growth (revised Figure 2E, Figure 2—figure supplement 2D).

Moreover, intracranial tumors arising in *Rag2*^-/-^ mice was much smaller when receiving transferred CD8^+^ T cells, but not CD4^+^ T cells (revised Figure 2H-I, Figure 2—figure supplement 2E).

To further evaluate the involvement of CD8^+^ T cells on controlling metastasis in metastatic locations or affecting dissemination of cancer cells, we performed the following experiments in the revised manuscript:

1) We first evaluated whether the involvement of CD8^+^ T cells on controlling metastasis in affecting dissemination of cancer cells. We analyzed the disseminated tumor cells (DTC) from peripheral blood of C57BL/6 mice injected with luciferase-overexpressed (luc-) EO771 parental or LM cells. The DTCs in the blood were defined as CD45^-^luciferase^+^ cells by flow cytometry (revised Figure 4—figure supplement 4A). We found that the percentages of DTCs were not significantly different between two groups (revised Figure 4—figure supplement 4B-C). Moreover, we injected EO771-luc LM cells into *Rag2*^-/-^ mice and transferred CD8^+^ T cells later. We found that CD8^+^ T cell transfusion did not influence the percentages of DTCs (revised Figure 4—figure supplement 4D), suggesting that CD8^+^ T cells did not affect the dissemination of cancer cells.

2) We observed that in the meninge, the number of CD8^+^ T cells of mice injected with EO771 LM-phenotype cells (LM-CD8^+^ T cells) was much lower than the one in mice injected with parental cells (Parental-CD8^+^ T cells) (revised Figure 4A-B, Figure 4—figure supplement 4E). Then, we further investigated the reason for the drop of CD8^+^ T cell count in the meninges. The tumor-specific CD8^+^ T cells were primed by DCs in dCLNs, migrated to meninges, and recognized and lysed tumor cells (19). Therefore, we evaluated the number of CD8^+^ T cells in dCLNs of mice injected with LM cells and parental cells. Interestingly, we found that the absolute count of LM-CD8^+^ T cells in dCLNs was significantly lower than Parental-CD8^+^ T cells (Figure 4C-D, Figure 4—figure supplement 4F). In addition, purified LM- and Parental- CD8^+^ T cells were labeled with CFSE and transferred into the tail vein of recipient mice. Two-photon live imaging showed that LM-CD8^+^ T cells migrated much less to the meninges in vivo, compared with Parental- CD8^+^ T cells (revised Figure 5A, Figure 5—figure supplement 5A). These results suggested that CD8^+^ T cell trafficking to meninges is impaired under leptomeningeal metastasis.

Taken together, we conclude that CD8^+^ T cells did not affect the dissemination process. T cell trafficking to the metastatic locations, however, was impaired, which led to acceleration of leptomeningeal metastasis. Nevertheless, the preclinical murine leptomeningeal metastasis model was established on the hematogenous dissemination through intracarotid artery injection, rather than the hematogenous spread from the primary lesion, which is different from the metastatic spread from human patients. Therefore, whether CD8^+^ T cells influence the dissemination of tumor cells in patients’ needs further study.

Although the message is interesting, I do suggest the authors to address immunology parts, especially T cell activation and the contribution of CD8 T cells on hampering leptomeningeal metastasis, in more details. Moreover, a direct examination the role of VLA4 expression in T cells on hampering metastasis is needed.

We thank the reviewer for the constructive suggestions. In summary, we have addressed this reviewer’s concerns as follow:

1) We have performed more convincing experiments to strengthen the conclusion that tumor-specific T cells activate in dCLNs and rewritten the immunology parts of Figure 3. We’ve demonstrated that tumor antigen-presented DCs and tumor antigen-specific CD8^+^ T cells were detected in tumor draining dCLNs, rather than in spleen or non-draining inguinal LNs (revised Figure 3A-B, Figure 3—figure supplement 3B-C). Moreover, DCs isolated from dCLNs of mice injected with EO771-OVA cells could specifically activated naive OT-1 CD8^+^ T cells in vitro (revised Figure 3C-E, Figure 3—figure supplement 3D-E)*.* Furthermore, we performed the surgical ligation of the lymphatic afferent to the dCLNs in mice to validate the generation of antigen-specific CD8^+^ T cells in dCLNs (revised Figure 3F-I, Figure 3—figure supplement 3F-H).

2) We have examined the contribution of VLA-4 in T cell adhesion to plate-bound VCAM-1-Ig. In addition, we employed an in vitro blood-brain barrier model to assess the transmigration of T cells (revised Figure 5G, H).

3) We’ve injected LM-phenotype or parental tumor cells in wildtype and *Trp53*^-/-^ mice and found that targeting p53 in CD8^+^ T cells can upregulate the expression of VLA-4 (revised Figure 6A) and enhanced their trafficking ability to meninges (revised Figure 6—figure supplement 6C-D), leading to effective tumor control (revised Figure 6—figure supplement 6E-F). Moreover, we’ve rewritten the Discussion sections to clarify the contribution of p53 in T cells and provide the novelty of our study.

4) We’ve analyzed the DTCs in the peripheral blood of the mice and found that CD8^+^ T cell transfusion did not influence the presence of DTCs (revised Figure 4—figure supplement 4A-D). Two-photon live imaging suggested that CD8^+^ T cell trafficking to meninges is impaired under leptomeningeal metastasis. These data implied that CD8^+^ T cells did not affect the dissemination process, and T cell trafficking to the metastatic locations was impaired, which led to acceleration of leptomeningeal metastasis.

Reviewer #2:In the submitted manuscript, Li et al. propose an interesting hypothesis that meningeal CD8^+^ T cells enter senescence in a p53-dependent manner in leptomeningeal metastasis (LM). While the observations are intriguing, the manuscript is not prepared carefully and lacks information critical for rigor and reproducibility. The use of a single mouse model severely limits the generalizability of the claims. Moreover, the mouse experiments are underpowered in all cases. The immunological observations are not internally consistent. Finally, the clinical validation materials are derived from what they call 'brain metastasis' (parenchymal metastasis), a completely different entity from leptomeningeal metastases, and these findings should not be generalized to LM.

We thank the reviewer for the constructive suggestions. We have revised the manuscript as follow:

1) We have adopted two more mouse models to prove the results: breast cancer cells 4T1 (revised Figure 1—figure supplement 1C-F, Figure 2—figure supplement 2A-B, Figure 5—figure supplement 5C) and murine non-small cell lung cancer cells LLC (Figure 1—figure supplement 1G-J, Figure 4—figure supplement 4I-K).

2) We’ve performed new experiments to further support the conclusion that dCLNs can facilitate the generation of tumor antigen-specific T cells. We detected the tumor antigen-presented DCs and tumor antigen-specific CD8^+^ T cells in tumor draining dCLNs, rather than in spleen or non-draining inguinal LNs (revised Figure 3A-B, Figure 3—figure supplement 3B-C). Moreover, DCs isolated from dCLNs of mice injected with EO771-OVA cells could specifically activated naive OT-1 CD8^+^ T cells in vitro (revised Figure 3C-E, Figure 3—figure supplement 3D-E)*.* Furthermore, we performed the surgical ligation of the lymphatic afferent to the dCLNs in mice to validate the generation of antigen-specific CD8^+^ T cells in dCLNs (revised Figure 3F-I, Figure 3—figure supplement 3F-H).

3) The cancer patients that we have included were all with leptomeningeal metastasis validated by MRI to validate our conclusion (Figure 7A).

4) We have compared the percentages and absolute numbers of immune cells in the CSF and meninges of mice injected with PBS, parental and LM cells. Details are presented as below:

Major Points (A-E):A. Many of the claims and conclusions must be reconsidered as the primary observations are derived from a single cell line. This is not sufficient to make generalized conclusions that are stated in the Results section, given the lack of similar observations in human breast cancer LM patients. The community standard for such claims is at least two cell lines (and corresponding models); three would provide even greater certainty.

We thank the reviewer for the suggestions and confirmed our results by using two more cell lines: 4T1 and LLC cells: breast cancer cells 4T1 (Figure 1—figure supplement 1C-F, Figure 2—figure supplement 2A-B, Figure 5—figure supplement 5C) and murine non-small cell lung cancer cells LLC (Figure 1—figure supplement 1G-J, Figure 4—figure supplement 4I-K).

The model itself is also inadequately characterized:– Quantification of disease burden is not consistent throughout the manuscript: The authors state that they quantified LM tumor volume from MRI experiments using the formula commonly used to quantify subcutaneous tumors: length x width2 x 0.52. Given that LM forms sheet-like structure on the surface of the brain, rather than a sphere, this formula is not appropriate. Later in the manuscript, they use (2D) area rather than volume. This discrepancy must be addressed.

In the previous manuscript, we used 2D area to quantify the metastatic lesions in slices(20,21). In the revised manuscript, we performed T1 weighted Gd-enhanced MRI to confirm leptomeningeal metastases (Figure 1D, Figure 1—figure supplement 1F and 1J) as described before (22). Higher sensitivity of Gd-enhanced MRI can identify tumor by differential contrast uptake in meningeal layers (dura-arachnoid or pia-arachnoid) to offer a definitive diagnosis over CT and CSF examination. Intensities and distribution of Gd contrast in MRI scans indicate the presence of leptomeningeal metastasis. We have rewritten this part in the Methods (Line 480-483, Page 17).

– New models of metastasis generated through iterative in vivo selection are typically introduced with the corresponding transcriptome. This is a great service to the larger metastasis community. It would also go a long way toward overcoming the use of a single cell line in the manuscript. Is it possible for the authors to characterize the bulk transcriptome of Parental, Intermediate and LeptoM cells?

We thank the reviewer for the suggestions. As the reviewer suggested, we characterized the bulk transcriptome of parental, intermediate and LM-phenotype cells by RNA sequencing (RNA-seq). Analysis of the transcriptome of these cell lines demonstrated that the gene expression profiles of the parental, intermediate and LM-phenotype cell lines segregate independently by principal component analysis(Figure 1H). Thus, each LM-phenotype population was both phenotypically and transcriptomally distinct from its matched parental and inter population.

– Tumor cells in Figure 2B are not really apparent. Also, different brain areas seem to be showed. Images and insets from the same anatomic locations would be helpful.

We have reperformed the experiments and supplied more apparent images and insets from the same anatomic locations in Figure 2E and Figure 2—figure supplement 2D.

B. Certain of the Immunologial observations presented are either internally inconsistent or are inconsistent with clinical data.– Authors use gP as a control for OVA experiments. Cancer cells overexpressing gB (human?) were introduced into wild type mice with adoptively transferred OT-I CD8^+^ cells, then dcLN CD8^+^ cells were isolated and challenged in vitro with corresponding peptides. No response is observed when CD8^+^ cells from E0771 gB mice are cultured with the peptides. In figure 2F the authors show that dcLN CD8^+^ cells from E0771 primed mice are able to control the growth of cancer and provide survival benefit, i.e. the cells in dcLNs recognize E0771 cancer cells to some extent. How would authors explain and address this discrepancy – the lack of gB specific CD8 T cells? Can authors isolate cells from E0771 WT mice and expose them to E0771 WT lysate to estimate the proportion of cancer-specific cells?

We thank the reviewer for the constructive suggestion. We have realized the experimental design before was insufficient to prove the hypothesis, and we performed additional experiments to support our conclusion that dCLN can facilitate generation of tumor antigen-specific T cells. The details are as below:

1) It’s well documented that the generation of tumor antigen-specific CD8^+^ T cells depends on effective tumor antigen presentation by dendritic cells (DCs) in lymph nodes (LNs), which is the fundamental step that launches T cell response against tumor (1,2). Thus, we firstly examined the peptide/MHC class I complex on the cell surface of DCs of mice bearing leptomeningeal metastasis, which indicated the tumor antigen presentation by DCs in LNs. Since deep cervical lymph nodes (dCLNs) could communicate with meningeal lymphatics directly, dCLNs were considered as the tumor draining LNs and inguinal LNs were taken as nondraining LNs. We inoculated EO771 breast tumor cells with ectopic expression of chicken ovalbumin (OVA) into the cisterna magna of C57BL/6 mice and isolated the CD11c^+^ DCs in dCLNs, inguinal LNs and spleen 7 days later. We found that OVA peptide/MHC class I complex, SIINFEKL/H2-Kb, was only detected on CD11c^+^ DCs in dCLNs, but not on the ones in inguinal LNs or the spleen (revised Figure 3A, Figure 3—figure supplement 3B).

2) Further, to investigate whether dCLNs can give rise to tumor-specific CD8^+^ T cells during tumor dissemination to leptomeningeal space, we injected PBS, EO771 cells (EO771 WT), EO771-OVA and EO771 with an irrelevant antigen glycoprotein B (EO771-gB) into C57BL/6 mice via the cisterna magna (revised Figure 3B). We used H-2Kb-OVA/SIINFEKL pentamer staining to detect OVA-specific CD8^+^ T cells in dCLN after tumor inoculation and observed that only DCs from mice receiving EO771-OVA inoculation could generate OVA antigen-specific T cells (revised Figure 3B, Figure 3—figure supplement 3C). Further, DCs were subsequently isolated from draining dCLNs, and then co-cultured with naive OT-1 CD8^+^ T cells (revised Figure 3C). Analysis of OT-1 T cell priming revealed that only DCs from mice receiving EO771-OVA inoculation could expand OT-1 T cells (revised Figure 3D, Figure 3—figure supplement 3D) and induce a significant increase in IFN-γ production of OT-1 T cells (revised Figure 3E, Figure 3—figure supplement 3E).

3) In addition, surgical ligation of the lymphatic afferent to the dCLNs which could impair the antigen presentation in LNs was performed before EO771-OVA tumor cell inoculation (revised Figure 3F). Determined by the expression of SIINFEKL/H2-Kb complex on the surface of CD11c^+^ DCs, antigen processing (revised Figure 3G, Figure 3—figure supplement 3F) was impaired in the mice receiving surgical ligation. In addition, the proliferation (revised Figure 3H, Figure 3—figure supplement 3G) and IFN-γ production (revised Figure 3I, Figure 3—figure supplement 3H) of OT-1 T cells primed by DCs were diminished.

– In Figure 4, the authors are showing drop in meningeal T cell numbers in the presence of cancer. LM is known to induce pleocytosis, this would not be reflected in relative numbers and thus absolute CD3 and CD8 numbers should be provided. What immune cell types are taking over meninges in the presence of cancer?

We thank the reviewer for the suggestions. In the previous manuscript, we found that the number of CD8^+^ T cells in the meninges of mice injected with LM cells was much lower than the one in mice injected with parental cells. In addition, two-photon live imaging showed that T cells isolated from mice with LM migrated much less to the meninges in vivo*,* compared with those from mice injected with parental cells. These results suggested that CD8^+^ T cell trafficking to meninges is impaired under leptomeningeal metastasis.

As the reviewer mentioned, cerebral spinal fluid (CSF) showed pleocytosis in patients with leptomeningeal metastases(23,24). As previously described, "Pleocytosis" refers to increased CSF nucleated cell (NC) count beyond reference range (typically >5 NC/µl), which occurs in the patients with infections (61.3%), miscellaneous (12.7%), vascular (9.7%), neurodegenerative (7%), neoplastic (5%), and inflammatory disease (4.2%) (25). In details, lymphocytes were the predominant population in CSF of patients with lymphomatous meningitis (LyM) and leptomeningeal carcinomatosis (LC), while LC patients often presented an increased rate of monocytes and the polymorphonuclear cells in CSF(24). In our previous manuscript, we only compared the percentages of meningeal T cells in mice injected with parental and LM cells, but not the T cell number in CSF. Since T cell floating in the CSF were able to reattach to the leptomeninges and invade the parenchyma, we compared the absolute numbers of different immune cell types in the leptomeninges and CSF of mice injected with PBS, parental and LM cells, respectively (revised Figure 2A-B). We found that the meninges of mice injected with parental and LM cells showed more CD45^+^ leukocytes, compared with the mice injected with PBS(revised Figure 2A). We analyzed the absolute numbers of meninge-infiltrating immune cells, including T cells, monocytes, microglia, myeloid cells and neutrophils(26). Interestingly, we observed that only T cells were markedly decreased in leptomeningeal metastasis (revised Figure 2B, Figure 2—figure supplement 2A), especially CD3^+^CD8^+^ T cells (revised Figure 4A-B). The main immune cell types in the and meninges of mice injected with LM cells were monocytes (27.75±29.95 cells), CD4^+^ T cells (185±47.67 cells) and microglia(40±24.59 cells).

– Authors claim that in CD8^+^ T cells in P53 deficient mice do not downregulate VLA4 and likely do not undergo senescence. Can authors reproduce the results from Figure 5 in P53 deficient mice? Does cancer grow less in these mice? BLI signal should be quantified.

As the reviewer suggested, we have reproduced the experiments to further illustrate the contribution of p53 in senescence(15). We injected LM-phenotype or parental tumor cells in wild type (WT) and *Trp53*^-/-^ mice to investigate whether p53 targeting can improve the metastasis control. In *Trp53*^-/-^ mice, the senescence of CD8^+^ T cells was inhibited as a result of p53 deficiency (revised Figure 6—figure supplement 6A-B), leading to upregulation of VLA-4 (revised Figure 6A) and enhanced trafficking ability to meninges (revised Figure 6—figure supplement 6C-D). Therefore, tumor growth was inhibited in P53 deficient mice determined by BLI signal (revised Figure 6—figure supplement 6E-F).

C. Setting aside the use of a single mouse model, there are serious issues with the work's rigor and reproducibility:– Some of the antibody dilutions (1:00) seem incorrect.

We apologize for the mistakes and have corrected them in the revised manuscript in Page 18 line 491, line 496.

– For Figure 5 source data, full membrane scans should be provided. The images that are shown are still somewhat cropped.

The images was taken by G:Box Chemi XT4. We’ve already provided the full membrane scans without editing and the shown images haven’t been cropped in Figure 4 source data 1.

– Why were the LeptoM cells established in Rag2-/- mice (line 306)?

We apologize for the misunderstanding caused. LM-phenotype cell was established in wild type C57BL/6 mice, but not *Rag2^-/-^* mice (revised Figure 1A). Instead, we inoculated LM-phenotype cells in *Rag2^-/-^* mice to investigate the role of T cells in leptomeningeal metastasis (revised Figure 2C-D). We have revised the legends and Methods.

– Authors claim they performed whole mount stainings of meninges, but such images are nowhere to be found.

We apologize for not referring to whole mount staining of meninges in previous Figure 4D. Images of previous Figure 4D were parts of whole mount meninges. In the previous studies, images of whole mount staining were also presented in a similar way (27,28). In the revised manuscript, we supplied the entire image of whole mount meninges and the inside lymphatic vessels were indicated by Lyve-1(red) (revised Figure 5C).

– In methods, authors say that the data are representative of at least three independent experiments with 4-6 mice per experiment. Most experiments in the submitted manuscript show only 5, sometimes 4 animals. Why did the authors not include the other two experiments, i.e. majority of animals? Please, provide data from all performed experiments as dot plots (not bar plots).

We are sorry for the mistake. The number of events and information about the statistical details and methods are indicated in the relevant figure legends. As suggested, we have provided the data from all performed experiments as dot plots (Figure 1C, 2A-B, 2D, 4B, 4D, 4I, 5F-H, 6C-D, Figure 1—figure supplement 1A-B, 1D, 1H, Figure 2—figure supplement 2C-E, Figure 3—figure supplement 3B-H, Figure 4—figure supplement 4C-D, 4G-H, 4K, Figure 5—figure supplement 5A-E, Figure 6—figure supplement 6B, 6D, 6F).

– The actual BLI quantification should be shown in main figures.

As suggested, the actual BLI quantification was shown in the main figures adjacent to the images accordingly (Figure 1B-C, 2C-D, 5E-F; Figure 1—figure supplement 1C-D, 1G-H, Figure 2—figure supplement 2B-C, Figure 6—figure supplement 6E-F).

– In Figure 2, authors state that tumor growth was monitored by BLI, but show pictures of brains ex vivo. It should be clearly stated in the figure legends what signal and at what time point was quantified.

We apologize for the unclarity. Tumor growth was monitored by BLI once a week. Mice were sacrificed when leptomeningeal metastases were detected by BLI, or the clinical signs of brain metastasis, including primary central nervous system disturbances, weight loss, and behavioral abnormalities, were shown. The images of Figure 2A were ex vivo bioluminescence imaging at day 28 post-injection when mice were sacrificed. In the revised manuscript, we have provided the time point of the detection in the legends (Page 33 line 972).

– Additional positive and negative controls for the P53 ChIP in Figure 6 should be provided. It would be also helpful to align the mutant Itga4 promoter sequence below the WT one in Figure 6B.

We thank the reviewer for this suggestion. As the reviewer suggested, we re-performed ChIP with additional positive and negative controls. Mutant *Itga* promoter sequence was shown in Figure 6B. p53 binding to exon 4 of the *Gapdh* gene and C*dkn1a* were used as negative and positive controls, respectively. We observed an average 4.26-fold enrichment was obtained with anti-p53 in the CD8^+^ T cells of mice injected with EO771 LM-phenotype cells, as compared to ChIP with a control immunoglobulin G (IgG). By contrast, an average 1.03-fold enrichment was found in T cells injected with parental cells(revised Figure 6D).

D. The methods section is incomplete and does not provide sufficient detail for interested readers to reproduce the experiments, and/or for adequate assessment of the experiments presented. Methods need to be carefully checked and all the lacking information regarding reagents and their use should be provided. In particular:

As suggested, in the revised manuscript, we have provided sufficient detail of the methods, amended the mistakes and carefully checked the method section.

– Details related to senescence analysis with ImageJ are not provided.

As suggested, we added method for quantification of SA-β-gal^+^ T cells in Page 24 line 676-678. For quantification of SA-β-gal-positive cells, images were randomly taken at 40× magnification (BX-53, Olympus) and then manually counted with ImageJ.

– It is unclear how were the in vivo treatments with antibodies performed.

As suggested, the concentration and the route of administration were defined in Page 18 line 513-518.

*“*in vivo administration of antibodies

Depleting antibodies to CD4 [GK1.5] and CD8 [53.6.72] were administered by intraperitoneal injection (0.25 mg/mouse) on day 1, 4 and 6. Blocking antibody to VLA-4[PS/2] was administered by intraperitoneal injection (0.25 mg/ mouse) on day 0, 1, 2 since the day of tumor cell injection. Rat-anti-mouse IgG (Cat.No. BE0090) was used as control antibody. All antibodies were obtained from BioXCell.”

– Gating strategy for tissues used in this manuscript should be shown.

As suggested, all gating strategies were shown. Gating strategies for Figure 2A-B, 3A, 4A, 4C, 7B and Figure 4—figure supplement 4B was shown in Figure 2—figure supplement 2A, Figure 3—figure supplement 3A, Figure 4—figure supplement 4E, 4F, Figure 7—figure supplement 7A and Figure 4—figure supplement 4A, respectively.

– Authors state that they purified CD4^+^ T cells and confirmed their purity with flow cytometry, but no anti-CD4 antibody is mentioned in the methods.

As suggested, we added the lacking information in the method section(Page 19 line 546).

– OT-I mice are not mentioned in methods.

As suggested, we added the lacking information in the method section(Page 15 line 430-431).

OT-1 mice on a fully C57BL/6 background were obtained from Shanghai Model Organisms Center Inc (Shanghai, China). All mice were bred and maintained in the specific-pathogen-free (SPF) animal facility of the Animal Experiment Center of Sun Yat-Sen University.

– Were the transgenic mice maintained as homozygous colonies? This statement should be added into the corresponding methods paragraph.

*Rag2*^-/-^ and OT-1 mice were maintained as homozygous colonies, and *Trp53*^-/-^ mice were obtained from Shanghai Nanfang Moshi Company. As suggested, we added the lacking information in the method section(Page 15 line 429-431).

*Trp53*^-/-^ mice, OT-1 mice on a fully C57BL/6 background were obtained from Shanghai Model Organisms Center Inc (Shanghai, China). All mice were bred and maintained in the specific-pathogen-free (SPF) animal facility of the Animal Experiment Center of Sun Yat-Sen University.

– Numerous viral vectors (luciferase, ovalbumin, glycoprotein B) are employed; the source and backbone are unclear.

As suggested, we added the lacking information in the method section(Page16 line 445-448).

EO771 cells were transduced with the viral vectors of ovalbumin (EO771-OVA, GenePharma) or glycoprotein B (EO771-gB, Guangzhou Ige Biotechnology) (multiplicity of infection[MOI] of 10) overnight at 37℃ with 5 μg/mL polybrene (GenePharma). The established cells were selected by 2 μg/mL puromycin(Σ).

– There is no source, concentration, length of incubation mentioned for several reagents (OVA and gB proteins, CFSE etc.)..

As suggested, we added the lacking information in the method section(Page 20 line 563-565).

OT-1 T cells were labeled with 0.5μM CFSE (Thermo Fisher, Cat.lot. C34554) for 15 min at 37℃.

– How were the CSF pellets preserved and stored?

As suggested, we added the lacking information in the method section(Page 25 line 718-720).

CSF sample were centrifuged at 450g for 8 min, and the cells of the CSF sediment were collected and preserved in cryovials with the 1 mL of freezing medium (90% FBS supplemented with 10%DMSO) in liquid nitrogen for future flow cytometry analysis.

E. The clinical samples do not provide validation for the overall hypothesis:– Although the authors analyzed an impressive number of clinical samples (Figure 7), the metastatic samples seem to be derived from "brain metastases" and not LM. Given this limitation, the proposed observations from a single mouse model cannot be reflected to human LM.

We thank the reviewer for raising an important question. In the clinical samples that we collected, there were 145 patients with non-malignant neurological diseases and 45 patients with leptomeningeal involvements which were confirmed by MRI as shown in Figure 7A, including 6 cases of breast cancer, 35 cases of lung cancer, and 4 cases of gastrointestinal cancer.

– An inverse correlation is demonstrated in Figure D; a negative Pearson's correlation coefficient is expected.

*R*-squared was previously provided in Figure 7D. We have corrected it in the revised manuscript.

– Gating strategy for human samples has to be shown.

As suggested, gating strategy for Figure 7B-C was shown in Figure 7—figure supplement 7A.